# Correlating activities and defects in (photo) electrocatalysts using in-situ multi-modal microscopic imaging

Camilo A. Mesa [1,2,3,12], Michael Sachs [1,4,5], Ernest Pastor [2,6], Nicolas Gauriot [7], Alice J. Merryweather [7], Miguel A. Gomez-Gonzalez[8], Konstantin Ignatyev [8], Sixto Giménez[2], Akshay Rao [7], James R. Durrant [1,9] & Raj Pandya [7,10,11] ✉

Photo(electro)catalysts use sunlight to drive chemical reactions such as water splitting. A major factor limiting photocatalyst development is physicochemical heterogeneity which leads to spatially dependent reactivity. To link structure and function in such systems, simultaneous probing of the electrochemical environment at microscopic length scales and a broad range of timescales (ns to s) is required. Here, we address this challenge by developing and applying in-*situ* (optical) microscopies to map and correlate local electrochemical activity, with hole lifetimes, oxygen vacancy concentrations and photoelectrode crystal structure. Using this multi-modal approach, we study prototypical hematite ($\alpha$-Fe$_2$O$_3$) photoelectrodes. We demonstrate that regions of $\alpha$-Fe$_2$O$_3$, adjacent to microstructural cracks have a better photoelectrochemical response and reduced back electron recombination due to an optimal oxygen vacancy concentration, with the film thickness and extended light exposure also influencing local activity. Our work highlights the importance of microscopic mapping to understand activity, in even seemingly homogeneous photoelectrodes.

The (photo)electrocatalytic splitting of water to hydrogen and oxygen is a key reaction for renewable energy conversion and storage. Metal-oxide semiconductor photoanodes such as TiO$_2$, WO$_3$ and $\alpha$-Fe$_2$O$_3$, continue to attract great interest for water splitting, despite having large overpotentials[1–4], due to their high chemical stability, low cost and relative ease for scale[5]. The efficiency of catalytic reactions in these materials is primarily determined by their ability to generate and efficiently separate charge carriers, and the kinetics of water oxidation when these charges accumulate at the material surface. It is the kinetic mismatch between these charge generation and accumulation processes, in particular the bulk carrier lifetimes (i.e., recombination and transport) and catalysis (i.e., hole transfer to water), over a range of

[1]Department of Chemistry and Centre for Processable Electronics, Imperial College London, London W12 0BZ, United Kingdom. [2]Institute of Advanced Materials (INAM) Universitat Jaume I, 12006 Castelló, Spain. [3]Sociedad de Doctores e Investigadores de Colombia, Grupo de Investigación y Desarrollo en Ciencia Tecnología e Innovación - BioGRID, Bogotá 111011, Colombia. [4]SLAC National Accelerator Laboratory, Menlo Park, CA, USA. [5]PULSE Institute, SLAC National Accelerator Laboratory, Stanford University, Stanford, CA, USA. [6]CNRS, Univ Rennes, IPR (Institut de Physique de Rennes) - UMR 6251, F-35000 Rennes, France. [7]Cavendish Laboratory, University of Cambridge, J.J. Thomson Avenue, CB3 0HE Cambridge, UK. [8]Diamond Light Source Ltd., Harwell Science and Innovation Campus, Didcot, Oxfordshire OX11 0DE, United Kingdom. [9]Department of Materials Science and Engineering, Swansea University, Swansea SA2 7AX, United Kingdom. [10]Laboratoire Kastler Brossel, ENS-Université PSL, CNRS, Sorbonne Université, Collège de France, 24 rue Lhomond, 75005 Paris, France. [11]Department of Chemistry, University of Warwick, Coventry CV4 7AL, United Kingdom. [12]Present address: Catalan Institute of Nanoscience and Nanotechnology (ICN2), CSIC, Barcelona Institute of Science and Technology, UAB Campus, 08193 Bellaterra, Barcelona, Spain. ✉e-mail: rp558@cam.ac.uk

microscopic length scales and reaction timescales[6,7], that limits the overall reaction efficiency in these metal-oxide photoanodes[8–10].

Structural and chemical disorder generated during the material synthesis[11–13] can strongly impact the photogenerated hole dynamics in metal-oxide semiconductors. This disorder can be broadly split into two categories: (i) micro-structural defects such as cracks and grain boundaries[14–17] and (ii) 0D point defects, which includes impurity doping or composition heterogeneity such as oxygen vacancies[18,19]. Both are known to play important roles in the catalytic properties of photoelectrodes[18] but investigating their function in-*situ* with the necessary simultaneous chemical, spatial and temporal resolution has remained challenging.

Techniques such as scanning electrochemical microscopy (SECM) have been used to visualise (with nanometre resolution) the distribution of dopants on photoelectrode surfaces and study water splitting in heterojunction photoelectrodes[20–24]. However, SECM, and its variants, are challenging to apply on rough electrode surfaces and generally provide limited temporal information on carrier dynamics[25,26]. Alternatively, a combination of scanning electron microscopy[27,28], surface photovoltage microscopy[15,16,29,30] and Kelvin probe measurements have been used to link defects and activity[31] and visualise charge transfer at a range of photocatalytic surfaces and interfaces, but in-situ studies remain challenging[32,33]. Optical (super-resolution) fluorescence microscopies[34–38] can somewhat alleviate the problem of in-situ probing and have unveiled the influence of inter-facet junctions[39,40], defect sites and nanoparticle structure on photocatalytic activity in a host of systems (2D materials[35,41], oxides and zeolites[17]). But the reliance on indirect exogenous fluorescent probes makes this method difficult to apply to a wide range of photocatalytic systems.

A need hence remains to develop and apply, inexpensive, non-destructive and label-free tools for investigation of (photo)electro-catalysts as they perform water splitting. Furthermore, given that numerous types of disorder can influence water oxidation, an approach with chemical, temporal and spatial resolution must be taken such that a complete picture can be obtained.

In this work, we develop a multi-modal microscopy setup encompassing ns to ms transient reflection imaging[42,43], Raman imaging, and local (spectro)electrochemistry, to map sub-micron electrochemical dynamics in photoelectrocatalysts. We apply this tool to image the local photocurrent (and fill-factor) across different regions of an α-Fe$_2$O$_3$ photoelectrode and correlate this with both the population dynamics of photogenerated holes, the local oxygen vacancy concentration, and structure. We find that, even in relatively homogeneous thin-film photoelectrodes, microscopic defects strongly influence hole dynamics and local activity, particularly via the rate of unfavourable back electron recombination processes. We further develop our findings using in-situ X-Ray near-edge absorption microspectroscopy (μ-XANES) suggesting that under light and bias, disorder (oxygen vacancies) might be mobile in certain morphologies of α-Fe$_2$O$_3$. Our results emphasize the importance of moving beyond an ensemble approach to build structure-function relationships even for thin-film photocatalytic systems and indicate that some types of microscale disorder may in fact be beneficial for water oxidation in thin-film oxides.

## Results

### Correlated optical microscopy

In this work we study ~300 nm thick films of α-Fe$_2$O$_3$ (hematite) on conductive fluorine tin-doped oxide glass (FTO), prepared using a hydrothermal synthesis method (see Methods; films annealed at 500 °C)[44,45]. A modified electrochemical cell (Fig. 1) with optical access is used to perform microspectroscopy under bias and measure local photoelectrochemical activity. All the applied potentials are referenced to the reversible hydrogen electrode (RHE; $V_{RHE}$) and, unless otherwise indicated 1 M NaOH (aq; pH 13.6) was used as an electrolyte.

Photocurrent mapping as shown in the top panel of Fig. 2 builds a spatial picture of the catalytic activity, via the photocurrent generated when illuminating specific regions of the sample (at 532 nm). The photocurrent density is correlated with the hole lifetime using transient reflection microscopy (TRM). Here, a combination of a focussed pump pulse at 400 nm and a focussed continuous wave (CW) probe pulse at 638 nm allows monitoring of the time resolved photoinduced absorption (PIA) of holes[46]. In this way, we can measure the population dynamics of photogenerated holes with sub-micron spatial precision over 10 μm × 10 μm regions as a function of applied bias (Fig. 2, middle panel). Over the same regions we perform spectrally resolved, bias

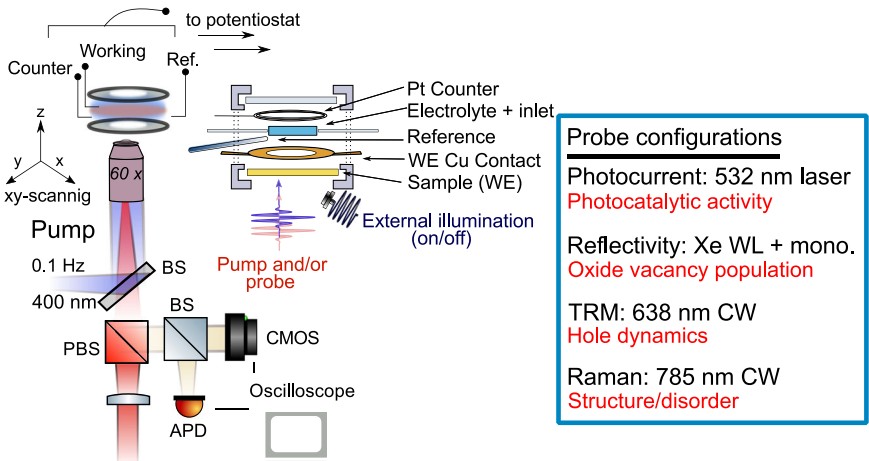

**Fig. 1 | Schematic of multi-modal correlated microscopy setup: Pump and probe sources are coupled into a home-built reflection microscope.** The pump (only used in transient reflection microscopy (TRM)) is a 0.1 Hz, 400 nm pulsed laser (5 ns pulse width), focussed to 440 ± 10 nm on the sample with a 60× 0.9 N.A. objective. The probe is varied depending on the experiment as detailed in the blue box (size range: 900 ± 10 nm to 1.3 ± 0.05 μm spot size; red text details the information obtained from each experiment). In the case of reflectivity measurements, the probe is wide-field (40 μm spot size). An 80:20 beam splitter, diverts signals to an avalanche photodiode (APD) for collection and camera for imaging. A 3-electrode electrochemical cell with optical access allows for application of potentials and collection of spatio-temporal optical signals. Unless stated illumination/light collection is always from the back side of the sample. An additional external illumination source is used for some Raman (400–550 nm) and photocurrent measurements (400–900 nm) as detailed in the text. In experiments with focussed pump/probe light, the sample is scanned to build 2D images.

## Local Photocatalytic Activity

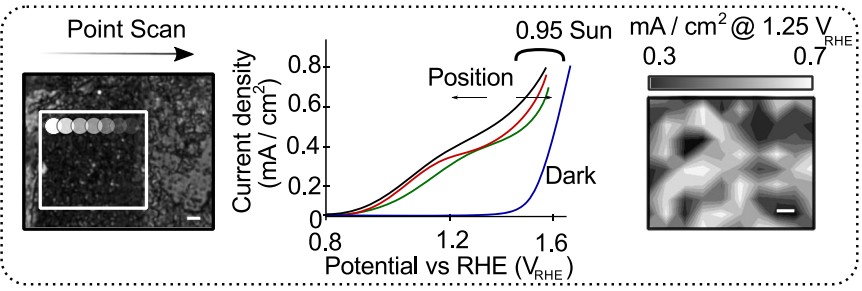

## Microscopic Hole Dynamics

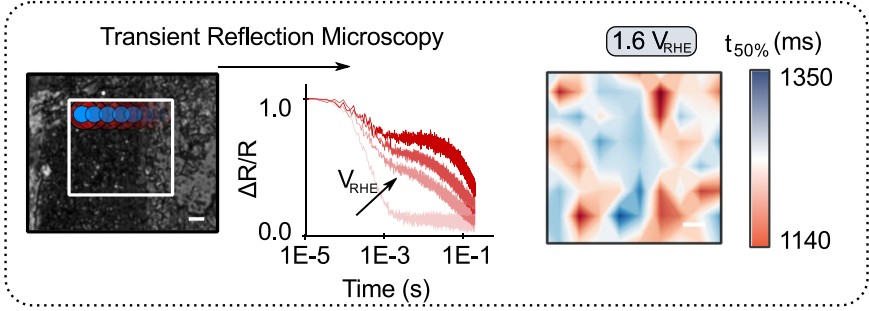

## Vibrational Structure & Oxide Vacancies

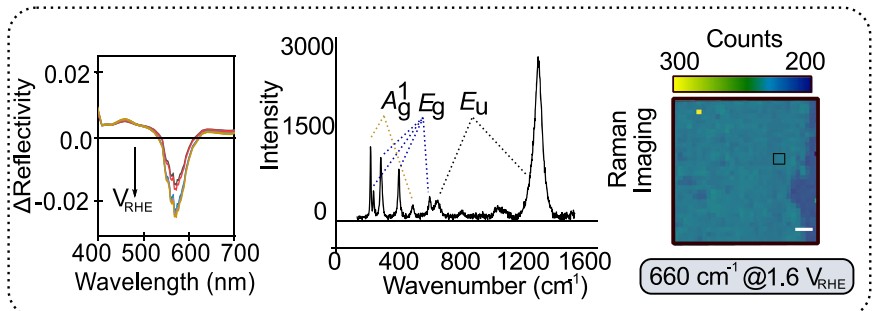

**Fig. 2 | In-situ simultaneous mapping of activity, hole dynamics and defect composition in a model (photo)electrocatalyst.** (Top) Scanning a focussed 532 nm light source (~1.3 µm FWHM; ~8 MW cm$^{-2}$ intensity at sample) allows the photocurrent density under illumination to be measured locally at different regions in the sample. The spatial photocurrent density is then plotted at 1.25 V vs RHE ($V_{RHE}$). A dark CV is shown in the middle panel (in blue) for reference. (Middle) Scanning of the sample allows transient reflection microscopy kinetics (ΔR/R) to be obtained at each spatial location. The kinetics from each TRM experiment are used to obtain the time taken for the signal to decay to half its initial value ($t_{50\%}$) as a function of bias and spatial location. (Bottom) Differential reflection spectrum as a function of bias (with respect to 0.5 $V_{RHE}$), shown for the region marked by the black box in the middle panel. Raman imaging (500 ± 10 nm spatial resolution) of a hematite photoelectrode measured using point scanning. The Raman spectrum of α-Fe$_2$O$_3$ is plotted in the bottom left with the various modes and their corresponding symmetries ($A^1_g$, $E_g$ and $E_u$) marked. Image displays the spatial intensity of the 660 cm$^{-1}$ mode at open circuit potential. These measurements give spatial insight into the concentration of oxygen vacancies. Scale bars on all images in the figure are 1 µm.

dependent, microscopic reflection imaging (bottom panel of Fig. 2). For materials like α-Fe$_2$O$_3$ it has been shown that the magnitude and bias sensitivity of the band-edge reflection spectrum correlates directly with the oxygen vacancy population[47,48] via intraband absorptions. Finally, Raman imaging (bottom panel of Fig. 2) provides insight into local chemical structure and its evolution under bias (and illumination). All optical measurements in our work are performed through the backside of the FTO substrate (see Methods section for further details on the exact illumination and biasing conditions for each measurement configuration).

**(Photo)electrocatalysis at microstructural breaks in the film**
Microstructural defects in (photo)electrocatalysts often occur in the form of breaks and cracks. In the hematite films studied here these are between 500 nm and 1500 nm in width (see Supplementary Note 1). We sub-categorise a prototypical 10 × 10 µm region containing cracks, as shown in the optical and SEM images in Fig. 3a, into three regions of

similar thickness: crack (termed CR; marked with a blue asterisk), adjacent-to-crack (termed ACR; marked with a green asterisk) and flat un-cracked film (termed FL; marked with an orange asterisk). Plotting the EDX spectra of the ACR and CR regions subtracted from the spectrum in FL regions (Fig. 3a right panel), shows that CR regions have a lower Fe and O content as compared to FL. Conversely, ACR regions are richer in Fe than FL, with slightly more O. Calculating the relative concentration of O compared to Fe (O:Fe; see Methods) from the EDX spectra, we find ACR regions are overall oxygen deficient[49,50] compared to FL (0.59 *versus* 0.65, respectively). However, we emphasise for such inhomogeneous samples, with light elements like O, the EDX values are only semi-quantitative and do not necessarily reflect only oxygen deficiency from vacancies (see Supplementary Note 1 and below). Indeed, because of the sub-micron length scales we work at, quantitatively determining the local concentration of oxygen (or other elements) remains challenging, even by methods such X-ray photo-electron spectroscopy. We also note that there will be some sub-300

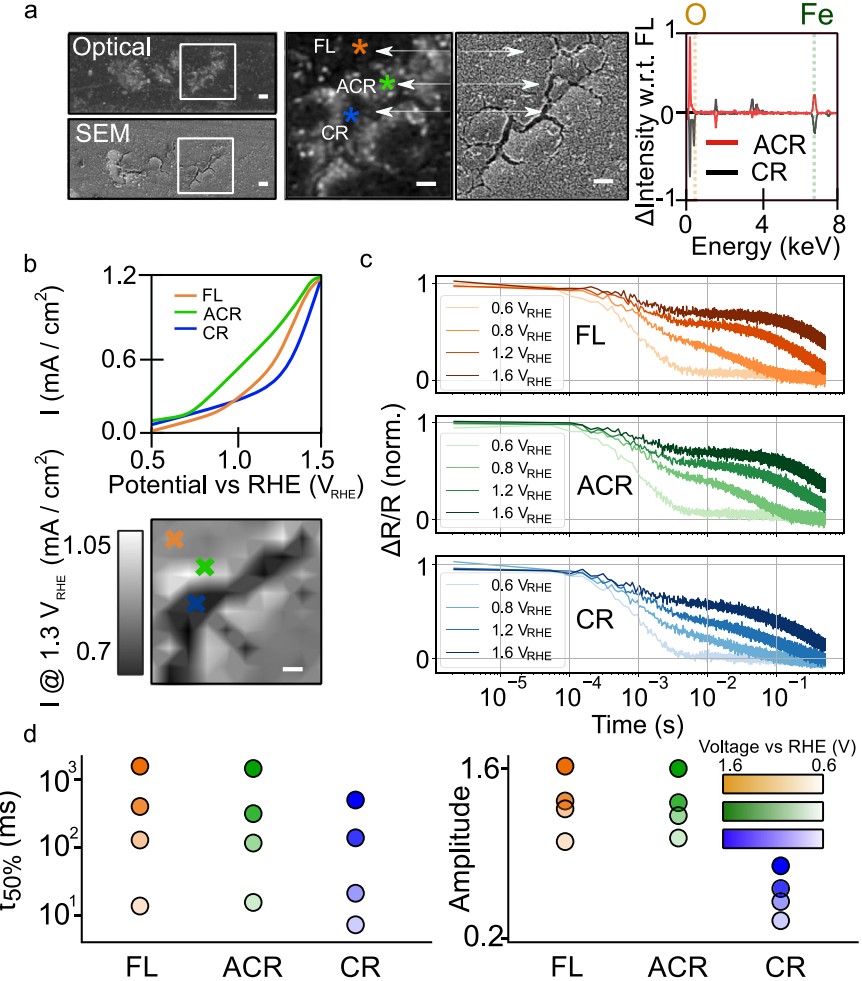

**Fig. 3 | OER reaction at and near cracks. a** Optical and SEM image of prototypical region of hematite photoanode displaying cracks (blue asterisk), regions of α-Fe₂O₃ adjacent to crack (green asterisk) and untextured film (orange asterisk). Far right panel shows EDX spectra from adjacent to crack (ACR; red) and crack (CR; black) regions subtracted from the EDX spectrum in film (FL) regions. Fe: peaks at 0.71 eV and 6.4 eV; O: peak at 0.51 eV; remaining peaks arise from trace impurities Si and Cl, and Sn from the FTO substrate underneath. **b** Photocurrent density map at 1.3 $V_{RHE}$ of regions shown in **a**. The orange, green and blue curves/crosses show CVs. **c** Kinetics of hole decay at 638 nm from FL, ACR and CR regions as a function of potential. **d** $t_{50\%}$ and amplitude of kinetics (see Supplementary Note 5) from FL, ACR and CR regions of sample. The error bars on values is not shown to ease visualisation, but there is a combined measurement and fitting error of ~10% on both $t_{50\%}$ and the amplitude. Scale bars are 1 µm in all images in the figure.

nm morphology present in our films (e.g., nanorods see Supplementary Note 2) which we cannot explore, due to the spatial resolution of the methods explored here.

To link the microscale composition of the catalyst with electrochemical activity we measure the photocurrent over the same area as shown in Fig. 3b (see Methods). Two salient features can be discerned from the photocurrent density map (plotted at 1.3 $V_{RHE}$) and I-V curves: (i) at the maximum potential of ~1.5 $V_{RHE}$ FL, CR and ACR regions produce a similar amount of photocurrent and (ii) the oxygen deficient ACR region exhibits a photocurrent onset ~200 mV earlier than the FL region (~0.7 $V_{RHE}$ *versus* ~0.9 $V_{RHE}$). Below potentials where the photocurrent plateaus, both water oxidation and back electron transfer reactions (BER) can occur. From the curves in the Fig. 3b the earlier photocurrent onset for ACR suggests BER is turned off more quickly in this region and ACR regions have a higher performance (i.e., fill factor; FF).

Interestingly, the current densities in Fig. 3b are slightly higher than might be expected based on other studies of hydrothermally synthesised α-Fe₂O₃ e.g., 1.2 mA cm⁻² at 1.5 $V_{RHE}$ here *vs* 0.7 to 1.1 mA cm⁻² at 1.5 $V_{RHE}$ in ref. 51. But the local current densities do match with macroscopic measurements of the sample (Supplementary Note 2), and have OER onset potentials that would be expected for

such defective films[44]. Nonetheless, to ensure our local I-V curves link to performance we can use the local current density, local absorption and theoretical current extractable from the sample under solar illumination (see Supplementary Note 3) to obtain a semi-quantitative estimate of both the spatially varying fill-factor as well as the incident photon to current efficiency (IPCE)[52]. The local fill factors vary between of 0.36 and 0.21, and the IPCEs from 22% to 10% (±4%), both in the order ACR > FL > CR, with a similar spatial distribution to the current density at 1.3 $V_{RHE}$ (Supplementary Note 3). We can compare our local IPCEs to those previously obtained in the literature (at 532 nm and 1.5 $V_{RHE}$). Interestingly, in ACR regions our IPCEs reach close to that of high-performance α-Fe₂O₃ prepared by chemical vapour deposition[52]. In other words, our defective film is in places comparable to an optimised sample. While the trends in FF and IPCE underscore the link between the local I-V curves, water oxidation efficiency and morphology we emphasise that our experiments do not allow us to comment on the OER mechanism (see Supplementary Note 3 for further discussion). We note our trends reported above are also seen in other regions of the film near cracks (see Supplementary Note 4).

To understand these performance differences further, we study the dynamics of photogenerated holes[53], over the same spatial region as imaged in Fig. 3a using transient reflection microscopy (see

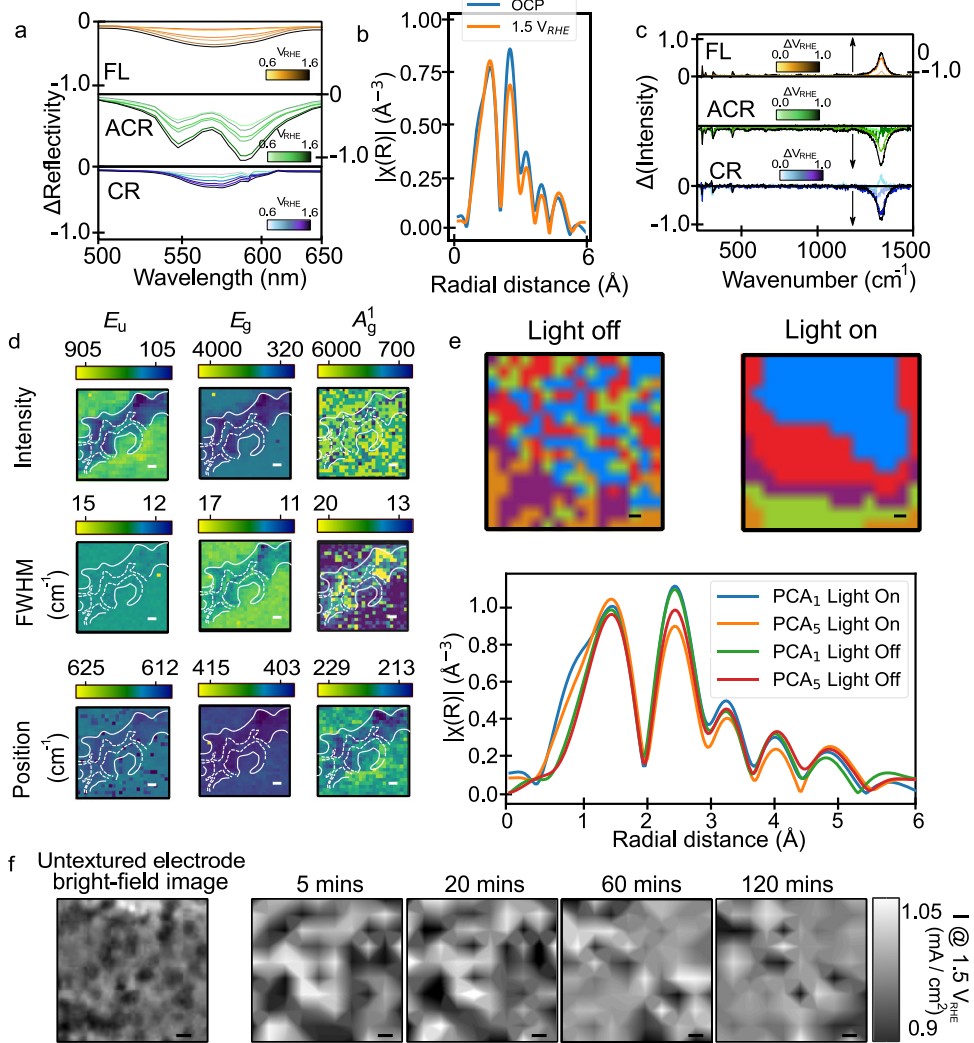

**Fig. 4 | Oxide vacancies and chemical structure at and near cracks. a** Differential spectral reflectivity (with respect to reflection spectrum at 0.5 $V_{RHE}$) as a function of bias from the FL (oranges), ACR (greens) and CR (blues) regions marked in panel **a** of Fig. 3. The differential reflectivity becomes more negative with bias for all regions showing a double negative peaked feature for ACR regions and a single negative peak for CR and FL regions. **b** R-space plot of the Fe K-edge XANES spectra of the hematite ($\alpha$-Fe$_2$O$_3$) electrode in the dark at open circuit potential (i.e, no applied potential, blue trace) and after applying 1.5 V vs RHE (orange trace). **c** Differential Raman spectra (with respect to 0.6 $V_{RHE}$) from region shown in Fig. 3a. All scale bars are 1 μm and panels show a 10 × 10 μm region. **d** Image of intensity, FWHM and position of indicated Raman modes at OCP. White dotted and solid lines

mark the boundaries of the CR and ACR regions, respectively. **e** (Top) Principal component analysis (PCA) of XANES absorption maps of a 20 × 20 μm hematite surface under dark (left) and under illumination (right) at 1.5 V vs RHE. The XANES absorption maps are found to contain 5 principle components denoted by the blue, red green, purple and orange colours in the maps. (Bottom) R-space plot of components 1 and 5 from of PCA analysis. Scale bar in images is 2 μm. **f** (Left) Bright-field image of homogeneous FL-like region of haematite photoelectrode. (Right panels). Local photocurrent map measured at 1.5 $V_{RHE}$ after light-soaking for the times indicated above the images. Some segregation in the photocurrent density can be observed after >60 min. Scale bar in all images is 1 μm.

Introduction and Methods). Although several equations can be used to describe the decay of the transient ΔR/R signal, we focus on using $t_{50\%}$ (half-life) to avoid multiple fitting parameters. The rate of decay represents both bulk electron-hole recombination and slower decay processes of surface localised holes (water oxidation and back electron hole recombination[54,55] at mild oxidative potentials). In Fig. 3c we plot the ΔR/R decay at ACR, CR and FL regions. For all three regions $t_{50\%}$ increases with bias, consistent with previous observations[54,56]. However, as the potential is raised $t_{50\%}$ follows the trend ACR ≈ FL > CR. At strong biases (~1.6 $V_{RHE}$) $t_{50\%}$ is ~1110 ms in ACR and FL, but ~940 ms in CR, whereas at lower biases (0.8 $V_{RHE}$), where BER quickens the hole decay, $t_{50\%}$, is ~115 ms and ~86 ms for the ACR/FL and CR regions, respectively (~10% error on values). The results in Fig. 3c suggest that more potential is required to separate charges and turn off BER in the crack regions, in agreement with the higher potential needed for

generation of photocurrent. We note that if we examine the magnitude of the non-normalised kinetics as shown in Fig. 3d (and Supplementary Note 5) we find that the ΔR/R signal amplitude follows the order ACR > FL > CR. This quantity will be a measure of yield of surface holes, suggesting there are fewer holes generated in the cracks.

To test the above hypotheses further, we can use microscopic reflectivity measurements to examine the concentration of oxygen vacancies which have been suggested to play a key role in back electron transfer reactions[54,57–59]. The spatially dependent reflectivity is measured in Fig. 4a as a function of a bias (with respect to the reflectivity at 0.5 $V_{RHE}$) in the FL, CR and ACR regions. In FL and CR there is a single negative peak in the (non-normalised) differential reflection spectrum centred at ~570 nm. In ACR there are two peaks, one at 550 nm and another at 575 nm (spectral resolution ~2 nm). The differential reflectivity signal at 570 nm has been previously assigned to

the generation of ionised oxygen vacancies (OVs), i.e., electron removal, corresponding to space charge layer formation, although reactive surface states may also play a role in this signal. Such OVs are typically associated with an excess of electrons resulting in the reduction of $Fe^{3+}$ to $Fe^{2+}$ in α-$Fe_2O_3$. The larger magnitude of the differential reflectivity in ACR regions as compared to FL/CR suggests that the vacancy concentration is higher.

The presence of the second differential reflectivity peak at 550 nm in ACR regions is challenging to assign. Comparison of our spectra to those in the literature, alongside electroabsorption and polarised reflectivity measurements (Supplementary Note 6), allows us to rule out its origin being from higher oxidation Fe species, Fe=O/Fe-OH/*O and specific nanostructuring[60–62]. Performing broadband fs-transient absorption spectroscopy on the same α-$Fe_2O_3$ samples (at open circuit potential; OCP) reveals that the decay kinetics are uniform in the spectral region associated with the peaks and that the additional peak could arise from a state that is present in the absence of bias (Supplementary Note 6). Combining this observation with literature calculations[63] and absorption measurements of epitaxial haematite[64] suggests that the higher energy transition is related to a second type of hole species in the valence band of the semiconductor, i.e., another Fe *d-d* transition with similar electronic/chemical properties to the lower energy transition. However, further experimental and theoretical work is required for full assignment.

In other metal oxides, such as $WO_3$, an intermediate oxygen vacancy concentration ([OV]) of 2 to 3% has been found to be optimal for water splitting[48]. This is because oxygen vacancies aid local charge separation and release charge carriers to the surrounding lattice improving conductivity. The higher [OV] in ACR regions, as compared to CR (and even FL), alongside the larger fill factor and IPCE and greater $t_{50\%}$, suggests that the oxygen vacancy concentration sits in an intermediate regime where the negative trapping effects are balanced against the charge separation properties they imbue. Finally, we note that OVs can also influence surface reaction kinetics, acting as coordination sites or sites to accumulate surface holes at. However, recent studies, on hematite in a range of morphologies, have revealed that the reaction *mechanism* does not change, regardless of the filling/density of OVs[44]. This implies that there is a more dominant role of OVs in recombination losses rather than on the surface reaction kinetics. Nonetheless, surface reactions linked to OVs may play some role in the hole dynamics detailed in Fig. 4c, and further work using e.g., micro-scale transient photocurrent mapping is required to fully establish the extent of this. We however note further methodological developments in terms of working at low excitation densities and in rough morphologies[65] are required before such measurements can reliably be performed at the microscale (see Supplementary Note 3).

To further clarify the properties of oxygen vacancies under bias (in the dark), we perform XANES measurements on homogeneous regions of the hematite film. Sun and co-workers[66] have reported that the relative intensity of the Fourier transformed XANES data in R-space, for the Fe-O bond, can be indicative of the density of oxygen vacancies in hematite films. Figure 4b shows the R-space data, from the Fe K-edge XANES spectrum of our haematite films. At first glance the Fe-O bond appears insensitive to the applied bias, suggesting the density of OVs remain constant as function of bias (see Supplementary Note 7). However, the intensity of the Fe-Fe bond (at 2.48 Å) does drop under strong positive bias (1.5 $V_{RHE}$), with a slight shift towards positive energies in the absorption edge (see Supplementary Note 7). This supports our notion (and the data in Fig. 4a) that there is a general increase in disorder and possibly the oxidation of $Fe^{2+}$, linked to OVs, to $Fe^{3+}$, under bias.

We can use Raman imaging (Fig. 4c, d; raw spectra in Supplementary Note 8) to link the above kinetic and thermodynamic observations to local structure. The key Raman active modes in α-$Fe_2O_3$ are of $A^1_g$ and $E_g$ symmetry and are centred at 224 $cm^{-1}$ and 493 $cm^{-1}$ ($A^1_g$)

and 244 $cm^{-1}$, 293 $cm^{-1}$, 411 $cm^{-1}$ and 610 $cm^{-1}$ ($E_g$). Structurally these correspond to the Fe-O bond stretch and an O-O symmetric stretch. The oxygen vacancy properties have been linked to the width and intensity of $E_g$ symmetry peaks[67,68] (specifically the intensity ratio between 293 $cm^{-1}$ and 411 $cm^{-1}$ modes[69,70], although there remains some debate as to which Raman features correlate most robustly with OVs/disorder in haematite). The 660 $cm^{-1}$ and 1320 $cm^{-1}$ (double phonon/double magnon; $E_u$ symmetry) mode intensities relate to the presence of FeOOH[71,72].

In the dark (off-resonant excitation) there is a similar intensity (same light blue colour) for the $E_g$ (and $E_u$) modes in the FL and CR regions as shown in the top row of Fig. 4d (all measured at OCP), whereas in ACR the intensity of these modes is reduced. The $E_u$ and $E_g$ modes are also narrower in ACR than in FL/CR ($A^1_g$ modes are slightly broader). Overall these observations suggest that while ACR has a higher OV concentration based on the reflectivity (above), the disorder is lower[73,74] in this region with reduced FeOOH.

Under bias the Raman modes evolve in intensity and position. In ACR and CR, the $A^1_g$, $E_g$ and FeOOH modes all drop in intensity as the bias is raised, with $A^1_g$ and $E_g$ modes additionally shifting to higher wavenumbers (~1.3 $cm^{-1}$), as shown in the differential Raman spectra in Fig. 4c. Interestingly, in FL regions the intensity of all modes increases under bias with a shift in the $A^1_g$ and $E_g$ modes to lower frequencies (~1.1 $cm^{-1}$). These changes with bias are challenging to rationalise but suggest that there might be some reorganisation of vacancies, which change the length and electron density in Fe-O and O-O bonds, as the bias is raised. Indeed, previous work[75] has suggested that the water splitting mechanism in hematite involves the oxidation of a Fe(III)-O species to Fe(IV)=O, the formation of which would cause large lattice distortions (due to an accumulation of holes). We note that the Raman spectra do not report only on OVs, potentially explaining why the difference in behaviour between FL and CR/ACR is not observed in the differential reflectivity.

Repeating the above measurements under continuous illumination (400–550 nm; electrode pre-light soaked for 50 min) shows limited difference in the response except in the FL regions (Supplementary Note 8). Intriguingly, a clear increase in the intensity ratio between $E_g$ modes at 293 $cm^{-1}$ and 411 $cm^{-1}$, linked to the degree of disorder and oxygen vacancies in α-$Fe_2O_3$, is observed here[69,70]. This tentatively suggests that light can accentuate structural (disorder) and vacancy rearrangement in homogeneous regions of the sample under bias (see Supplementary Note 8 for further discussion).

In order to test the changes in disorder with potential and light further we perform spatially resolved XANES measurements on FL-like regions (2 μm spatial resolution; 20 × 20 μm region; not the same film area as above). To analyse the data, we first perform principal component analysis (PCA) of the XANES absorption map at an energy of 7125 eV, i.e., around the Fe-absorption edge. Following PCA, a cluster analysis is performed whereby those pixels exhibiting statistically similar spectra are grouped. Cluster analysis, shown in Fig. 4e, top left, highlights the presence of 5 clusters (blue, red, purple, green and orange data, respectively). However, in the dark there is no clear spatial dependence in the distribution of pixels or absorption values (see Supplementary Note 7 for further details). However interestingly, under illumination and 1.5 $V_{RHE}$ bias, the XANES absorption map (Supplementary Note 7) and PCA cluster map (Fig. 4e, top right) of the same region, exhibit spatial segregation of absorption values/clusters.

Fourier transforming the XANES data from the PCA analysis in R-space (Fig. 4e bottom panel) shows that the relative intensity of the Fe-O bond (at 1.5 Å) and Fe-Fe bond (at 2.49 Å) between clusters, i.e., different spatial positions, varies (comparing green and red, for light off, and blue and orange when the light is on). As discussed, these features could be linked to the OV population, suggesting that OVs are dynamically re-distributing under light and bias. This observation is also reinforced by measurement of photocurrent maps of FL-like

(homogenous) regions over extended time periods (Fig. 4f). Following >60 min of continuous white light exposure under bias (400–900 nm over the whole measurement area) we find a slight segregation of photocurrent values which again suggests structural rearrangements might be occurring (see also Supplementary Note 9). We note that as the measurement/illumination time in all other measurements detailed in this work is significantly shorter than the timescale over which we observe changes induced by bias and light, they will not be influenced by this effect unless stated (see Methods and Supplementary Note 8 for further details).

## The role of film thickness

The thickness of a photoanode is also an important optimisation parameter. Too thin and absorption will be poor, but too thick and mobile charge carriers generated within the film will not be able to reach the electrolyte/catalyst surface due to the modest hole diffusion lengths in metal oxides. To assess this at the microscale, we examine crack-free regions of the $\alpha$-Fe$_2$O$_3$ film which present several different thicknesses as shown in Fig. 5a. AFM scans reveal the three regions marked as T1, T2 and T3 with thicknesses of $310 \pm 5$ nm, $273 \pm 8$ nm and $145 \pm 5$ nm respectively (Fig. 5b). SEM/EDX suggests that there is a very weak decrease in the O:Fe content ratio from T3 to T1 (0.65 to 0.63; Supplementary Note 10). The content of Sn in T2 and T3 is ~3% lower than in T1 but given the 1–2 µm penetration depth of EDX such measurements may be dominated by the Sn signal from FTO. Nonetheless, we do not expect Sn diffusion into hematite layers to be significant.

Light absorption decays exponentially through the depth of the hematite film[28] following the Beer-Lambert law, becoming negligible below ~300 nm. Given this is the nominal thickness of our film (and the axial extent of the point spread function in our microscopy system is >500 nm), to appropriately compare photocurrent densities and absorption between regions of different thickness, the signals must be scaled by the light-attenuation inside the material (see Supplementary Note 11 for further details). Correcting for thickness variations, Fig. 5d shows a map of the photocurrent density across T1, T2 and T3. The photocurrent density in T3 at ~1.6 V$_{RHE}$ is slightly larger (~1.27 mA cm$^{-2}$) than T1/T2 (~1.20 mA cm$^{-2}$); 15% uncertainty on values. This trend is also observed in the local fill-factors and IPCE values (FF: 0.33 (T3) and 0.19 (T1); IPCE: 18% (T3) $vs$ 13% (T1)) which demonstrate the performance of T3 regions is on-par with high quality films[76]. Finally, the photocurrent onset follows the order T3 > T2 > T1, i.e., potential for photocurrent generation and back electron transfer is lowest in T3.

Examining the hole population decay we find in T2 and T3, $t_{50\%}$ is systematically larger for a given bias than in T1 (Fig. 6a), indicative of reduced BER in these regions (e.g., at 1.6 V$_{RHE}$ $t_{50\%}$ is ~1300 ms in T2/T3 and ~940 ms in T1). The magnitude of the $\Delta R/R$ signal also is ~1.4 times larger in T2/T3 as compared to T1 (see Supplementary Note 12 and below for corrections to $\Delta R/R$ signal magnitude), suggesting a larger hole population in-line with the higher photocurrent density at the saturating potential. Hole diffusion lengths are traditionally thought to be short[77] (10–100 nm) in metal oxide photocatalysts, hence despite recent reports of greater than 500 nm hole diffusion in Ti-doped hematite[78], we assume that the diffusion properties of T1, T2 and T3 remain similar and are not responsible for our observations. One factor however that may be important is that thicker films are more defective (see discussion below), thus generating an excess of OVs that might act as trapping centres increasing recombination processes.

To assess the defect properties, specifically in terms of oxygen vacancies, we again perform biased spectral reflectivity measurements. As the reflection contrast predominantly arises from the FTO/hematite interface, signals are not scaled by the sample thickness (but we note the same trends persist with/without scaling; see Supplementary Note 11). Figure 6b shows the differential reflectivity spectra

($versus$ 0.5 V$_{RHE}$) from T1, T2 and T3 on increasing the bias from 0.6 V$_{RHE}$ to 1.6 V$_{RHE}$. The spectra show a broad negative feature centred between 560 nm and 580 nm whose magnitude increases with bias. In T1 there is a larger change in the magnitude of the reflectivity, on increasing the bias, than T2/T3. Based on the previous assignment of features (for ACR/FL/CR) our observations suggest that in T1 there are a larger number of oxygen vacancies, as compared to T2/T3 (the ratio of reflectivity T1:T2 at 570 nm is ~3.5). Although we cannot quantitatively extract the vacancy concentration, the low FF and IPCE in T1 would suggest that [OV] is sufficiently high that vacancies are in a regime where they act as detrimental trap sites as opposed to facilitating conduction (i.e., being beneficial). This also explains the shorter hole lifetimes in T1 due to defect facilitated BER, along with the fact that electrons generated close to the surface need to travel further in thicker parts of the film.

Finally, to assess other forms of structural disorder we repeat Raman imaging (first in the dark) over the same region as shown in Fig. 5a. Several features can be noted in the Raman images in Fig. 6c: (i) the average intensity ratio $A^1_g$:$E_g$ is 1.34 in T1, whereas it is 1.02 and 0.93 in T2 and T3 respectively, (ii) the $A^1_g$ mode is shifted to lower frequencies in T2 and T3 and (iii) the FeOOH mode ($E_u$ symmetry) is also of higher intensity in T2 and T3 as compared to T1 suggesting there is some loss of symmetry and formation of tetragonal defects[72]. Interestingly, these differences between T1 and T2 qualitatively resemble those when hematite is thinned to the few-layer limit (hematene). In this case a high density of surface-active enhanced reaction sites and a locally modified charge around the adsorption sites[79,80] results in a greater photocatalytic activity, with similar behaviour also reported for BiVO$_4$[81]. The results in Fig. 6c suggest that similar structural modifications could be occurring in T2/T3 promoting water oxidation. On the application of bias, we observe a general increase in the Raman mode intensity for T1, T2 and T3 regions with a red-shift of $A^1_g$ and $E_g$ modes as was observed for FL regions (see Supplementary Note 8). Importantly in T3, which most closely resembles FL, the application of bias and illumination (same method as for ACR/CR/FL region) results in an increase in the ratio between Raman modes at 293 cm$^{-1}$ and 411 cm$^{-1}$. As previously discussed, this mode ratio reports on disorder (including OVs)[69,70], with our observations suggesting these might be redistributed by light and bias in T3.

Whilst we have focussed on mapping the influence of structural defects on (photo)catalytic activity, the above methods can also be used to examine the role of chemical impurities. In Supplementary Note 13 we examine the influence of native carbon containing impurities on hematite photocatalysis. We find while these impurities increase the hole-lifetimes, the poor absorption and reduced oxygen vacancies they imbue makes them detrimental for photocatalysis.

In summary we have demonstrated that microstructural and chemical defects strongly impact the local I-V fill-factor and IPCE (and hence efficiency) in an archetypal metal-oxide (photo)electrocatalyst, hematite. Our data suggests this is achieved by modulating both the rates of back electron recombination and influencing the oxygen vacancy concentration such that a balance between charge trapping and electronic conduction can be achieved. While the $\alpha$-Fe$_2$O$_3$ we study has an overall (macroscopic) poor photoelectrochemical activity, regions of $\alpha$-Fe$_2$O$_3$ adjacent to cracks and that are thinned relative to the bulk show higher performance than the native film, close to state-of-the-art haematite[52]. This suggests that certain types of imperfections may actually be beneficial for photoelectrocatalysis and could be used to boost performance if controllably engineered into a catalyst.

More generally our work highlights the need to move beyond macroscopic measurements. This necessarily requires the development of new measurement methods, but the all-optical approach demonstrated here is a first step in this direction being label-free, non-destructive and applicable in-situ. However, further developments will

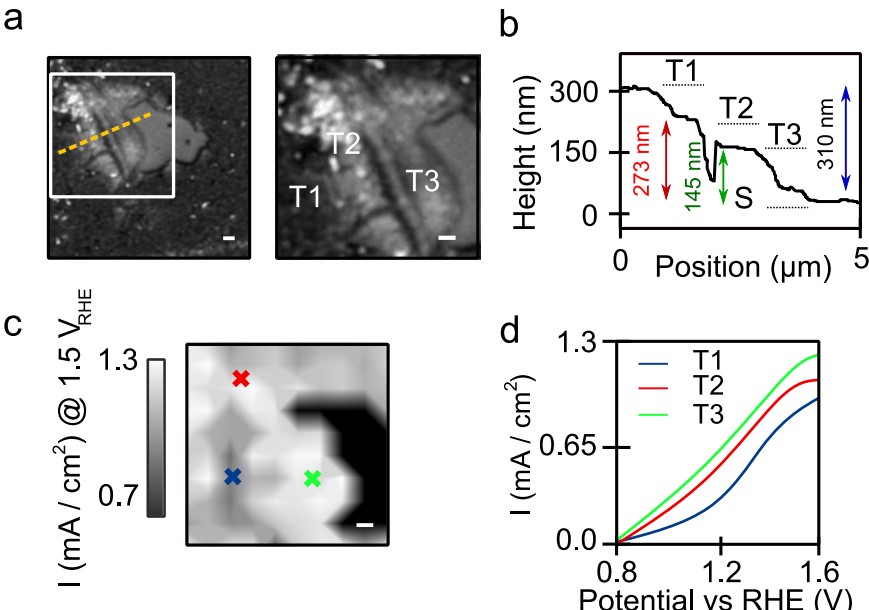

**Fig. 5 | Investigating the effect of photoanode thickness on activity. a** Optical image of hematite photoanode displaying multiple thicknesses. The orange dashed line indicates where the AFM line scan is taken. Right panel shows 10 μm² region investigated here (white box in left panel). **b** AFM topography profile with heights of three layers: T1, T2, T3 and substrate (S) marked. **c** Photocurrent density map at 1.5 $V_{RHE}$ from corresponding area in panel **a**. Blue, red and green crosses mark places where photocurrent traces/spectra are taken from in **d** and Fig. 6 (T1, T2 and T3 respectively). **d** Photocurrent density traces from areas marked in **c**. All scale bars are 1 μm in the figure and the panels in **a** (right side) and **c** show a 10 × 10 μm region.

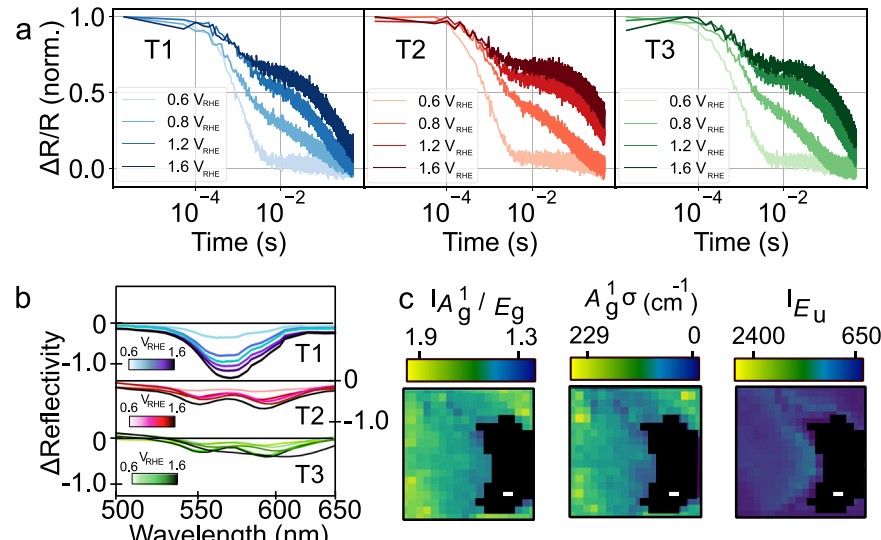

**Fig. 6 | Influence of thickness on hole dynamics, oxide vacancies and local structure. a** Kinetics of hole decay from T1, T2 and T3 regions (marked in Fig. 5a) as a function of bias. **b** Differential spectral reflectivity (with respect to reflection spectrum at 0.5 $V_{RHE}$) as a function of bias from T1, T2 and T3. **c** Ratio of Raman mode intensities $A^1_g/E_g$ (224 and 411 cm⁻¹ modes respectively) along with peak position (σ) for $A^1_g$ and intensity of the mode for $E_u$. Black region marks substrate. All scale bars are 1 μm in the figure and panel sizes show a 10 × 10 μm region.

be needed especially to track defect dynamics over extended time periods and link thin-film morphologies to the OER mechanism itself. This could include the integration of microscopic transient photocurrent mapping (and an appropriate analysis framework[65]), methods to quantify [OV] at sub-micron lengthscales, and using an LED for microscopic transient reflection measurements to better mimic solar irradiance[82]. The light-driven migration/segregation of disorder (potentially OVs) we observe is intriguing but characterising the (local) oxide vacancy diffusion coefficient and activation barrier will be instrumental in learning how to exploit this phenomenon. Finally, the pump-probe approach we employ can be used to track the dynamics of a wide range of carriers. For example, the transient reflectance signal can also be related to the thermal diffusivity[83,84]. By mapping this across an electrode[85] or thermoelectric material[86], local heat transport properties could be estimated and correlated with structure.

## Methods
### Sample preparation
Undoped α-Fe$_2$O$_3$ photoanodes were deposited by a hydrothermal method[44]. Briefly, irregular FeOOH was grown onto 25 mm × 25 mm

FTO coated glass from a solution containing 150 mM $FeCl_3$ and 1 M $NaNO_3$ at 100 °C for 1 h. After rinsing with DI water, the FeOOH was converted into $\alpha$-$Fe_2O_3$ by annealing at 500 °C for 1 h.

## Transient reflection microscopy

An EKSPLA NT340 tunable laser was used as the pump source. The pulse duration was 5 ns and the repetition rate was set to 0.1 Hz using a pulse picker. All experiments were performed at pump wavelengths between 380 nm and 510 nm; main text 400 nm. Three O.D. 1.0 absorptive filters (Thorlabs) were used to directly attenuate the output of the laser. The collimated pump was pinholed (75 μm pinhole), telescoped and directed using a 45 degree mirror to the objective (Olympus MPLAPON60X, 0.9 N.A) housed in a home-built microscope body. The probe was delivered by a 638 nm CW Coherent CUBE laser. The probe was pinholed (20 μm pinhole), passed through a 50:50 beam splitter (for collection) and then combined with the pump line using a dichroic mirror (DLP505 Thorlabs). The probe was focussed into the back focal plane of the objective using the same 200 mm lens telescoping the pump. Samples were placed in a 3-electrode cell (MM Raman ECFC 3.5 cm², 4.5 mL; redox.me) with the housing modified to accommodate for the objective working distance. The sample cell was built onto a x-y-z translation (Optosigma) stage to allow for fine focussing and sample scanning. The reflected probe was collected by the beam splitter and then passed through a second 80:20 beam splitter, with 20% of the widefield probe imaged onto a camera (FLIR Grasshopper 3) and the remainder focussed with a 50 mm lens onto an avalanche photodiode (APD; Thorlabs); long pass filters were used to reject any pump scatter. The time resolution of the setup was found to be ~30 ns. The output of the APD was connected to an oscilloscope (Keysight Infium 9000; ns to μs) or National Instruments (NI USB-6211) DAQ card (μs to ms) for measurement of the kinetics depending on the time range. Data were stitched together between the two time ranges in post processing. Between 1000 and 10,000 laser shots were recorded per spatial location to achieve sufficient signal to noise depending on the exact experiment. Potentials were applied with a Gamry Interface 5000E potentiostat. Home-built Python software was written for hardware interfacing and data acquisition. The probe intensity was set to ~100 μJ/cm² at the sample and the pump fluence varied to a maximum of ~500 μJ/cm² at the sample. Further details such as the wavelength and fluence dependence of signals is discussed in Supplementary Note 14. Measurements typically involved around 1 min of collection/illumination time at each spatial position, i.e., these measurements will not be influenced by any effects of long-term illumination.

## Micro-spectroelectrochemisty

The same microscope body and cell used for TRM measurements was used for micro-spectroelectrochemistry. No pump source was used, but instead a Xe white light (WL) lamp (200 W Newport) was passed through a monochromator (Bentham Instruments) and then coupled (Thorlabs RC02FC) into an optical fibre (500 μm diameter). The fibre output was focussed into the back focal plane of the same microscope objective as for TRM, to project the beam in the wide-field onto the sample. At a given bias, the monochromator wavelength was scanned in 2 nm steps and the reflection image measured on the camera (FLIR Grasshopper 3). Any beam splitters in the collection path were removed to collect the maximum reflected light. The bare FTO substrate was used to determine reference spectra which were subtracted from all data. Averaging over typically 25 spatial pixels allowed sufficient signal-to-noise to be obtained to determine the reflection spectra from the image stack. The reflectivity measurements require <5 min collection/illumination time across the whole water oxidation range and hence are not influenced by any effects of long-term illumination.

## Photo-electrochemical microscopy

Micro-photocurrent mapping was performed using the method outlined by Kozlowski et al.[87], Furtak et al.[88], Sambur et al.[38]. and Butler[89]. A 532 nm CW laser (Oxxius LCX-532S) was used to excite spots on the hematite anode through the backside of the FTO. The laser focus diameter was ~1.3 μm with a maximal power density of 8 MW cm⁻². The 532 nm laser excitation light was chopped at 36 Hz and a lock-in amplifier (Stanford Instruments) was used to detect the current from the excited region of the photoanode on top of the background of the rest of the photoanode via a potentiostat (Gamry Interface 5000E). The steady state photocurrent signal was averaged for 30 s at each potential. Overall, an I-V trace at a single position required <10 min of acquisition. The measured nanoampere photocurrents were normalised by the laser spot size to give a photocurrent density. The potential was stepped in 0.05 V increments from 0.5 $V_{RHE}$ to 1.6 $V_{RHE}$ with a smoothing spline interpolation between points to create the CV. For the light soaking experiments detailed in Fig. 4 the whole area of the electrode is soaked with white light (400–900 nm) for the time indicated. In this case maps were only measured at 1.5 $V_{RHE}$ for rapid acquisition (<5 min full map acquisition).

## Raman microscopy

Raman measurements were performed with a Renishaw inVia Raman microscope. Excitation was provided by a 785 nm laser which is which is off-resonant from the hematite absorption ensuring no additional photogenerated species are created. The Raman emission was collected by a Leica 50× objective (N.A. = 0.85) and dispersed by a 1800 lines mm⁻¹ grating. The samples were scanned with an x-y piezo stage to collect 20,000 spectra over 10 × 10 μm grid in 500 nm steps with 2 s dwell time at each spot. For potential and illumination dependent Raman measurements, the same configuration as for imaging was used with Raman spectra measured at each of the respective locations at a given potential. In this case spectra were acquired with a 2 s dwell time at each spot. For Raman measurements with illumination and bias the same conditions as above were used except the whole area of the electrode was illuminated for 50 mins before starting measurements with a white light source between 400–550 nm as to not interfere with the Raman detection.

## Atomic force microscopy

AFM measurements were done in tapping mode (Veeco Dimension 3100) at room temperature. The AFM cantilever was provided by MikroMasch. The tip radius was ~10 nm.

## SEM and EDX analysis

Secondary electron images were obtained using a MIRA3 TESCAN SEM system, at 5 kV. EDX spectra acquisition and analysis was performed using an Oxford Instruments AZtecEnergy X-Max$^N$ 80 EDX system, at 15 kV. The O:Fe ratio was calculated (within the Oxford Instruments AZtecEnergy Software) as $\frac{A_O}{\Sigma_j A_j}$, where $A_j$ is the adjusted intensity of a given element.

## XANES

Both point and mapping XANES experiments were collected at the Fe K edge using a Si(111) double-crystal monochromator and the Si drift detector with Xspress3 readout system in reflectance mode. The energy steps used were: 6962–7092 eV, 5 eV steps; 7092–7105 eV, 1 eV steps; 7105–7120 eV, 0.25 eV steps; 7120–7172 eV, 1 eV steps; 7172–7212 eV, 2 eV steps; 7212–7512 eV, 5 eV steps. A 0.05 mm Al filter was used to attenuate the fluorescence of the Fe Kα1 signal to avoid detector saturation. A sample stage was used for mapping 20 × 20 μm regions with a 1 μm step for both x and y axes. An Fe foil was used for the energy calibration and $FeCl_2$, $FeCl_3$, $Fe_2O_3$ and $Fe_3O_4$ were used as model compounds for $Fe^{2+}$ and $Fe^{3+}$ species. For μ-XANES measurements that involve illumination, the whole area of the electrode is

illuminated throughout the measurement whilst scanning the sample over the X-ray beam. Full µ-XANES maps took approximately 60 min per map.

XANES data analysis was carried out using Athena Demeter 0.9.26 XAS data processing software[90] and maps data were analysed by Mantis 2.3.02 software package[91].

## Data availability

The raw data that support the findings within this paper are available at https://doi.org/10.5281/zenodo.10866150[92].

## Code availability

Analysis codes are available from the corresponding author upon request.

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

## Acknowledgements

This work acknowledges funding from the Generalitat Valenciana program (APOSTD/2021/251, C.A.M.), MinCiencias Colombia (Fondo Nacional de financiamiento para la ciencia, la tecnología y la innovación "Francisco José de Caldas", call 848 de 2019, C.A.M.), the CNRS and the French Agence Nationale de la Recherche (ANR; grant ANR-22-CPJ2-0053-01, E.P.), a "la Caixa" Foundation Fellowshp (ID 100010434 (LCF/BQ/PR22/11920013), E.P.), the EPSRC via the Cambridge NanoDTC, (EP/L015978/1, A.J.M.) and grants (EP/M006360/1 and EP/W017091/1, A.R.), the Winton Program for the Physics of Sustainability (A.R. and R.P.) and Clare College, Cambridge (Junior Research Fellowship, R.P.). The authors (S.G. and C.A.M.) acknowledge support from project PID2020-116093RB-C41 funded through MCIN/AEI/10.13039/501100011033. This work was carried out with the support of Diamond Light Source, instrument I18 (proposal SP30381-1). R.P. additionally thanks Arjun Ashoka (Cambridge) for useful assistance and advice with building of the experimental setup and writing of the data acquisition code.

## Author contributions

C.A.M. measured and analysed X-ray data, interpreted the results and synthesised samples. M.S. interpreted the results. E.P. performed X-ray measurements and interpreted the results, N.B.G. performed Raman measurements. A.J.M. performed SEM and EDX measurements. M.A.G-G. interpreted the X-ray data. K.I. supervised X-ray measurements. S.G. supervised the work of C.A.M. A.R. supervised the work of A.J.M. and N.B.G. J.D. interpreted the results. R.P. designed the project, built and coded the transient optical microscope, performed measurements and analysed and interpreted the data. All authors contributed to the writing of the manuscript.

## Competing interests
The authors declare no competing interests.
