## [Peer Review File · Nature Communications]

Correlating activities and defects in (photo)electrocatalysts using in-situ multi-modal microscopic imagingREVIEWER COMMENTS

Reviewer #1 (Remarks to the Author):

Correlating activity and defects in (photo)electrocatalysts using in-situ transient optical microscopy
Mesa et al.

In this work Pandya and colleagues use in-situ optical microscopy as a tool for examining structure-dependent photocatalytic properties in the common photoanode material, α -Fe₂O₃ (hematite). While previously elements of in-situ optical microscopy including steady state and transient reflectance and photocatalytic activity have been established, here the authors extend the approach to include Raman based chemical imaging. This, in combination with complimentary techniques including SEM, XANES, and EDX spectra allowed for the correlation of structural features (including microscopic grain boundaries and oxygen vacancies) and catalytic behavior as extracted from a standard J-V curve. While the work presented here represents a step-forward in the spatiotemporal description of photoanode behavior, the lack of correlation to functional water oxidation current (as outlined in point 1) limits the scope of the results such that this manuscript is more appropriate for a field specific journal. Additionally, several technical and mechanistic points regarding the manuscript are listed below.

Comments:

(1) While the spatial mapping of photocurrent as a function of voltage is a significant development, previous work from the lead author (ref. 37) has already shown that strong variations in J-V curve behavior does not yield meaningful differences in water oxidation yield. As such, while the variation in onset voltage and qualitative fill-factor are noteworthy, they do not represent a full correlation to photocatalytic yield, limiting the utility of the spatial correlations discussed here.

(2) Building on this, either a direct or relative metric for fill factor would be helpful for discerning meaningful quantification of differences between structural morphologies and film thicknesses (as currently only qualitative trends are discussed).

(3) The Raman maps in Figure 4d are somewhat challenging to interpret and could be improved by adding a contour overlay of the either the optical or SEM image (in order to better match the flat, crack, and adjacent to crack regions).

(4) The raw (not differential) Raman spectra associated with Figure 4c should be included in the SI, as smaller features in the differential spectra are difficult to see in the Figure. For example, the peak position of the 610 cm⁻¹ mode is shown in the lower left panel of Figure 4d, but this peak is not discernible in the differential spectra included in panel c. Additionally, the authors should clarify which Raman maps were collected under illumination.

(5) Regarding the differential reflectance spectra (Figure 4a and 6b) the authors should assign the "two peaked structure" in the adjacent to crack and tier 2, tier 3, regions. For example, while the 550 nm feature could be a different type of oxygen vacancy (as discussed), what causes the redshift of the second feature from the assigned 570 nm towards 575-590 nm?

(6) The authors suggest that the oxygen vacancy within the adjacent to crack regions could reside in some type of "intermediate sweet spot" balancing charge separation and negative trapping effects. Could the OV concentration here be determined by the Fe:O ratios identified in the EDX spectra?

(7) The segregation of PCA1 and 5 under light illumination in Figure 4e is very interesting, however the lack of other data on this particular thin film makes it challenging to understand the origin of the spatially dependent features. The presentation of this data would be greatly improved if its chemical or potentiostat mapping were included here along with discussion of what could give rise to the spatial

dependence.

(8) The authors should check the scale bar for the Ag1 mode in Figure 6c, as it seems odd that the peak frequency would extend all the way to 0 cm^{-1} .

Reviewer #2 (Remarks to the Author):

Authors developed in-situ spatial imaging methodology (photocurrent, spectroelectrochemical reflection spectroscopy, transient reflection spectroscopy, Raman spectroscopy, and X-Ray near-edge absorption microspectroscopy) to reveal the spatial heterogeneity of reactivity and establish robust structure-function relationships. Using prototypical Fe₂O₃ photoanode, the correlation between oxygen vacancy concentration (crack and thickness) and photoelectrochemical performance is established. This work is novel and provides some insights for understanding activity and the impact of defects from microscopic mapping, which could be recommended for publication in Nature Communication after addressing the following comments.

Comment #1

Fe₂O₃ film was prepared by solvothermal (FeOOH) and following annealing (500 °C, Fe₂O₃). (1) This film wasn't activated by high temperature, and the high PEC performance is puzzling. A macroscopic PEC performance needs to be provided. (2) According to previous reports and Supplementary Note 1, this film usually shows irregular nanorod morphology and isn't dense film. The discussion about the impact of thickness needs to be careful.

Comment #2

Oxygen vacancy usually shows two functions, charge recombination and surface reaction. In manuscript, the authors only considered the impact of charge recombination, and ignored the effect on surface reaction kinetics. (1) In Figure 3, FL, ACR and CR show similar photocurrents at 1.5 VRHE, but the half lifetime ($t_{50\%}$) and amplitude from CR are obviously different from those of FL and ACR. These differences can't be understood by charge recombination. (2) In line 150-155, authors correlated the onset potential of photocurrent with back electron transfer reactions. This correlation is inaccurate, because severe recombination in bulk Fe₂O₃ photoanode can lead to large onset potential. The transient photocurrents and integral charges should be provided to compare back charge concentration in CR, ACR and FL.

Comment #3

In lines 184-188 and in Figure 4a, the peak assignment of 550 nm to a second type of hole species at dark is arbitrary. Because Figure 4a suggests that the lower potentials can induce the appearance of this peak, lower potentials can't generate high-energy holes. This peak need to be re-assigned.

Comment #4

(1) Imaging and spectra of Raman are only measured under OCP and potential-dependent conditions. In order to understand the impact of oxygen vacancy on PEC performance, the imaging and spectra of Raman under illumination-dependent conditions need to be provided. (2) In Figure 4a, the differential spectra demonstrated the order of oxygen vacancy concentration: ACR > FL \approx CR. However, the differential Raman spectra of CR and ACR in Figure 4c show a similarly negative trend, in contrast to that of FL. The different OER kinetics need to be considered and discussed.

Reviewer #3 (Remarks to the Author):

In this manuscript, Pandya et al. utilized a series of integrated microscopic mapping technologies to in

situ study the activity–structure relationship in a prototypical hematite (α -Fe₂O₃) photoanode materials. Using a combined characterization of photoelectrochemical imaging, transient reflection microscopy and in situ Raman measurement alongside an X-ray absorbance measurement, the authors unveiled a significant spatial heterogeneity in activity and structure of α -Fe₂O₃, and found a positive correlation between photoelectrochemical activity and oxygen vacancy concentration. Considering that bulk-level studies are usually carried out in this field to reveal a link, the method proposed herein holds promises in promoting the development of highly efficient photoanode materials. On the other hand, this manuscript is not well written regarding the logic structure and languages. Therefore, I recommend this work to be published after major revisions listed below.

1. The title of this manuscript “correlating activity and defects in (photo)electrocatalysts using in-situ transient optical microscopy” can’t completely describe the study carried out in this work. Maybe a title of “correlating the activities and defect concentration in (photo)electrocatalysts using integrated in-situ microscopic imaging technologies” will be better.

2. As for the steady-state reflectivity measurements of α -Fe₂O₃ shown in Figure 4a and 6b, the authors found that the most active region (ACR region or thinner area of T2/T3) possessed an interesting doublet signal, quite distinct from that of the lowly-active area. Have this phenomenon been reported in the case of α -Fe₂O₃ or other photoanode materials before? If using a transient reflection characterization, whether these double peaks will have same kinetics? How to completely understand the relationship between the electrochemical activity of α -Fe₂O₃ and such unique double peak signals? Maybe this is the most important findings obtained by integrated in-situ microscopic imaging technologies in this work. It will be better if the authors can strengthen this point.

3. It seems that Figure 5c has a mistake. The presented Figure 5c is a photocurrent mapping rather than an EDX mapping result of T1, T2 and T3. Moreover, there seems a contradiction, maybe. The authors claimed that T1, T2 and T3 area had same Fe:O ration from EDX mapping (seemingly no differences in oxygen vacancies in these three areas). But on the other hand, the authors proposed that T1 area has more oxygen vacancies according to the results that T1 had bigger reflectivity change (Figure 6b). This suggested that the reflectivity change isn’t only determined the concentration of oxygen vacancies, and also affected by the thinness of α -Fe₂O₃. In this sense, it is not solid to predict the oxygen vacancy concentration using steady-state reflectivity measurements (for example, Figure 4a).

4. Though integrated in situ microscopic mapping, the author indicated a positive correlation between photoelectrochemical activity and oxygen vacancy concentration in α -Fe₂O₃ and provided the possibilities to tune the oxygen vacancy concentration via generating microscale cracks or reducing the thickness of α -Fe₂O₃. Can the author further give a quantitative description between photoelectrochemical activities of α -Fe₂O₃ and oxygen vacancy concentration? This is hard to be accessed in general bulk-level study but can really show the power and promise of the integrated in situ microscopic mapping in this field.

5. Several other important literatures related to this manuscript should cited to help readers to better understand the background of this research (Fang et al., *Sci. Adv.* 2021, 7, eabj4452; Sadtler et. al., *J. Am. Chem. Soc.* 2021, 143, 11393–11403). And the most relevant work (i.e., Ref. 73, *Nature*, 2016, 530, 77–80) should be cited in the third paragraph of the “Main” part of this manuscript.

6. The readability and the language of this manuscript should be further improved, especially the part of in situ Raman and X-ray absorbance measurements. Maybe the authors should provide some schematic illustrations to correlate the Raman/X-ray characterization results with the photoelectrochemical activity of α -Fe₂O₃ to help readers to better understand these obtained structural information. Moreover, the manuscript should be checked more carefully. For instance, several obvious mistakes appear in the first paragraph of “(Photo)electrocatalysis at microstructural breaks in the film” section, such as “marked with red/green asterisk”, “peaks at 0.71 eV/6.4 eV and 0.51 eV”, “the Fe:O ratio in ACR regions”.

Response letter for: NCOMMS-23-27868: "**Correlating activity and defects in (photo)electrocatalysts using in-situ transient optical microscopy**"

We thank the reviewers for the careful evaluation of our manuscript, for their comments and help with improving the work. We have performed new experiments, analysis and reworkings of the text based on their feedback.

Below we address (in black) the specific points raised by each referee (in blue). Changes to the text are highlighted in black (bold) and red.

REVIEWER COMMENTS

Reviewer #1 (Remarks to the Author):

Correlating activity and defects in (photo)electrocatalysts using in-situ transient optical microscopy
Mesa et al.

In this work Pandya and colleagues use in-situ optical microscopy as a tool for examining structure-dependent photocatalytic properties in the common photoanode material, α -Fe₂O₃ (hematite). While previously elements of in-situ optical microscopy including steady state and transient reflectance and photocatalytic activity have been established, here the authors extend the approach to include Raman based chemical imaging. This, in combination with complimentary techniques including SEM, XANES, and EDX spectra allowed for the correlation of structural features (including microscopic grain boundaries and oxygen vacancies) and catalytic behavior as extracted from a standard J-V curve. While the work presented here represents a step-forward in the spatiotemporal description of photoanode behavior, the lack of correlation to functional water oxidation current (as outlined in point 1) limits the scope of the results such that this manuscript is more appropriate for a field specific journal. Additionally, several technical and mechanistic points regarding the manuscript are listed below.

We thank the reviewer for the time taken in evaluating our work and the constructive and helpful feedback. We are pleased to see the reviewer finds our work novel and interesting and agree while many of the techniques have been used before, combining them at the microscale is new. We do also believe we have pushed the methods significantly, for example performing Raman imaging with bias, different illumination conditions and high spatial resolution is not something we have seen done before. Similarly, transient reflection microscopy has, to the best of our knowledge, not been developed for the ns-ms time regime or with point-scanning of kinetics. These are non-trivial methodological developments. While there are limitations to the observations (*e.g.*, on how quantitative we can be on the oxygen vacancy concentration), that we more explicitly outline now, we believe our work does demonstrate a clear link between functional water oxidation efficiency and microstructure. Beyond developing a transferable methodology, we believe we have provided several new insights for water oxidation in thin film metal oxides. These include both fundamental insights such as on the microscopic light and bias induced oxygen vacancy migration, and more practical ones such as the fact that while in large fractions of samples the performance can be poor, in others it can rival that of the best photoanodes. This latter point implies that we might be able to engineer defects into photocatalysts to further boost their activity. Below we provide a detailed response to the

reviewer's questions. We hope in our answering, the reviewer finds our work improved and now suitable for publication in Nature Communications.

Comments:

(1) While the spatial mapping of photocurrent as a function of voltage is a significant development, previous work from the lead author (ref. 37) has already shown that strong variations in J-V curve behavior does not yield meaningful differences in water oxidation yield. As such, while the variation in onset voltage and qualitative fill-factor are noteworthy, they do not represent a full correlation to photocatalytic yield, limiting the utility of the spatial correlations discussed here.

We thank the reviewer for agreeing that the spatial mapping we show in our manuscript is a significant development. The reviewer raises an interesting point with regards to the lack of link between ensemble I-V behaviour, morphology and water oxidation yields and whether those same limitations apply when examining local behaviour. Firstly, we wish to note that in ref 37 (of the originally submitted text) the measurements were performed on photoelectrodes with different *nanoscale* textures and were performed at a *macroscopic* scale (cm lengthscales). This text concluded that differences in the I-V behaviour do not influence the water oxidation *mechanism of reaction*. The novelty of our work is we re-examine this question with the accessible microstructure. Furthermore, we are considering microscale point and structural defects cracks, thickness and impurities, these are quite different to the impact of nanomorphology considered in ref 37, where the rate determining step (RDS) of the OER in 4 different hematite photoanodes was studied, with different optical and electronic properties (including doping density and oxygen vacancies). In this text kinetic analyses revealed a 3rd order process with respect to surface hole density for all the samples studied, suggesting that the RDS is achieved when 3 photogenerated holes are accumulated at the reaction centre (see more details in *Nat. Chem.* 12 82-89 2020), regardless of the *nanostructure*. However, the conclusion that the water oxidation mechanism is not changed (and reported on by the I-V) is not established for our morphologies yet. Secondly, in ref 37 we show that the I-V curves do not necessarily give insight to the water oxidation **mechanism**, but this text does not say that the I-V curves do not provide insight into the efficiency or the overall yield of the process. We do generally believe (as we think the reviewer suggests) that the water oxidation mechanism should be the same in all regions, but further work will be needed to evidence this (for a complementary discussion, please see our discussion in question 4.2 of reviewer 2). While we believe that oxygen vacancies can participate in the surface reaction, as coordination sites or to accumulate surface holes, their role is most important in recombination losses. The main challenge in establishing the variation in the water oxidation **mechanism** with morphology is that transient photocurrent measurements are likely required to measure the rate-laws. Whilst we can perform these locally, interpreting them requires significant theoretical model development as traditionally such measurements rely on weak excitation and no optical scattering both of which would be challenging to achieve in our case (see detailed discussion in response to question 2 of reviewer 2).

As we demonstrate, more quantitatively with our new measurements below, microstructural and chemical defects strongly impact the 'local' I-V fill-factor (and hence efficiency) in haematite, an archetypal metal-oxide (photo)electrocatalyst. This is by modulating both the rate of back electron recombination and influencing the oxygen vacancy concentration such that a balance between charge trapping and electronic conduction can be achieved. We note that such kinetic competition would ultimately likely affect the population of surface holes affecting directly the OER kinetics, however, we hypothesise the nature of the OER mechanism

should be equivalent in these conditions following our previous observations in refs 10, 37 and 56 (of the originally submitted text). We do however believe this needs work to establish. **We have made note of these new points the reviewer raises in our revised text as follows:**

*‘While the trends in FF and IPCE underscore the link between the local I-V curves, water oxidation efficiency and morphology we emphasise that our experiments do not allow us to comment on the OER mechanism (see **Supplementary Note 3** for further discussion).*

...

However, recent studies, on hematite in a range of morphologies, have revealed that the reaction mechanism does not change, regardless of the filling/density of OVs⁴¹.

...

However, further developments will be needed especially to track defect dynamics over extended time periods and link thin-film morphologies to the OER mechanism itself.’

Finally, we wish to remark that assaying water oxidation efficiency from the I-V response is, at the ensemble level, something commonly done in the literature [e.g., *Chem. Sci.*, 2015, 6, 4009, *ACS Catal.* 2017, 7, 6, 4062–4069]. We hence, believe our approach is fair, if not with limitations. Furthermore, we emphasise that it is not only the spatial photocurrent mapping that is particularly novel, but the transient reflection microscopy and Raman imaging also. The former technique has thus far only been employed in the fs-ps regime [see: *J. Phys. Chem. C* 2023, 127, 30, 14557–14586] (which is arguably less relevant for the water oxidation mechanism, although as noted, all recombination processes will affect the ultimate density of surface holes and the OER kinetics) and has not been ever built in a point-by-point kinetic tracking manner. Similarly microscopic Raman imaging (or XANES imaging) also has not been performed with control over bias and illumination conditions. This latter set of measurements have allowed us to build detailed insights into oxide vacancy migration (see response to question 7). Generally, we believe the link between the spatial photocurrent maps we obtain (across water oxidation potentials) and carrier dynamics/structure is robust, and allows us to make practical conclusions such as the fact that engineering microdefects could be beneficial to thin-film metal oxide photoelectrocatalysis. The conclusion of our manuscript has been edited to reflect this as follows:

‘While the α -Fe₂O₃ we study has an overall (macroscopic) poor photoelectrochemical activity, regions of α -Fe₂O₃ adjacent to cracks and that are thinned relative to the bulk show higher performance than the native film, close to state-of-the-art haematite⁴⁹. This suggests that certain types of imperfections may actually be beneficial for photoelectrocatalysis and could be used to boost performance if controllably engineered into a catalyst.’

(2) Building on this, either a direct or relative metric for fill factor would be helpful for discerning meaningful quantification of differences between structural morphologies and film thicknesses (as currently only qualitative trends are discussed).

We agree with the reviewer that it is important to make our analysis as quantitative as possible. We have now estimated both local ‘fill factors’ (FF) as well as local incident photon to current efficiencies (IPCE) [*J. Am. Chem. Soc.* 2006, 128, 49, 15714–15721] for the film in the different spatial regions considered. For the fill factor, we define the thermodynamic oxidation

potential of water (1.23 V vs RHE) as the ‘short-circuit’ conditions. Given that the performance of a solar cell (**Figure R1a**) is normally characterised as the maximum power ($V_{\text{Max}} \times I_{\text{Max}}$) divided by the maximum power point, MPP ($V_{\text{OC}} \times I_{\text{SC}}$), we estimate equivalently the FF of our hematite photoelectrodes at 0 V of overpotential (η), *i.e.*, the thermodynamic potential of the OER (1.23 V vs RHE). Thus, the maximum photoelectrochemical power that can theoretically be achieved by our hematite electrodes is taken as $\eta \times I_{\text{Meas}}$ see **Figure R1b**, left panels. We note for η we take the onset voltage (when the current density exceeds 0.1 mA cm^{-2}), and for I_{Meas} we use the current density at 1.23V. Whereas the power that actually is achieved is $\eta_{\text{Onset}} \times I_{\text{Meas}}$, *i.e.*, the maximum in a plot power density vs voltage (**Figure R1b**, right panel). This method of calculation is supported by the work of Hodes [*J. Phys. Chem. Lett.* 2012, 3, 9, 1208–1213]

On the other hand, the IPCE at 1.23 V vs RHE can be determined by the ratio ($I_{\text{Meas}@1.23\text{V}}/I_{\text{Theor}@1.23\text{V}}$) between the measured photocurrent at 1.23 V vs RHE and the maximum achievable photocurrent density of our photoelectrode. This maximum photocurrent density is calculated using the absorption spectra of our hematite photoelectrode as follows:

- (i) The local absorption spectra at each point in the film are obtained by combining the reflection spectra measurements from **Figure’s 4 and 6** of the main text with transmission spectra measurements at the same location.
- (ii) The local absorption is multiplied by the total solar flux under AM1.5 solar irradiation (normalised solar spectrum). However, as in our measurements of photoelectrochemical activity we are using a laser at $532 \pm 15 \text{ nm}$, we only take the solar spectrum and absorption between 515 and 545 nm. In this way we can obtain the theoretically absorbed photon flux at each local point on the film examined.
- (iii) The absorbed energy is divided by the photon energy at each wavelength to give the maximum number of photo-generated electrons and holes.
- (iv) Using the above value and the integration time at each point in the local photocurrent measurements we can obtain the maximum possible photocurrent at each spatial location.
- (v) Comparing the above theoretical current density with the photocurrent density measured at the local water oxidation potential allows us to obtain a local FF/IPCE for each location in the film.

The above procedure is summarised in **Figure R2**. Although this analysis is simplified *e.g.*, does not account for dark currents it allows a semi-quantitative understanding to be obtained. Following the above procedure, we find a theoretical maximum current for our films of $3.07 \pm 0.1 \text{ mA cm}^{-2}$.

Based on the above value, examining the CR, ACR and FL regions as above we find the local FF values vary between of 0.36 and 0.21 in the order $\text{ACR} > \text{FL} > \text{CR}$ (**Figure R3a**). For T1, T2 and T3 the FF ranges between 0.19 (T1) and 0.33 (T3) (**Figure R3b**); errors are 0.06. The IPCE varies between 10% and 22% in the order $\text{ACR} > \text{FL} > \text{CR}$ (**Figure R3c**). For T1, T2 and T3 the FF/IPCE ranges between 13% (T1) and 18% (T3) (**Figure R3d**); errors are $\pm 4\%$. Interestingly the trend in FF/IPCE of $\text{T3} > \text{T2} > \text{T1}$ is more evident than for the photocurrent, suggesting indeed that subtle variations in the film thickness and associated vacancies can play a role in the water oxidation kinetics.

Figure R1: **a.** Schematic of how performance is estimated for solar cells. **b.** Outline of how we quantitatively estimate the fill factor of our haematite photoelectrodes. To estimate the theoretical power density that can be achieved we can take a plot of the overpotential vs current density from which we estimate the onset overpotential (when the current density exceeds 0.1 mA cm^{-2}). We can also determine the short circuit current from the point at which the voltage vs RHE reaches 1.23 V. Multiplying these together gives us the maximum power performance. Multiplying the current density and voltage and plotting this vs overpotential then allow us to estimate $|P|_{real}$ which is the maximum power actually achieved. From these two quantities a fill factor can be estimated.

Figure R2: Ensemble absorption spectrum and cumulative sum of theoretical current density as a function of wavelength based on film absorption and solar spectrum. The dotted line shows the region of the cumulative current density (and film absorbance) that contributes in the wavelength range we illuminate our electrodes in.

Figure R3: a-b. Map of fill factor (FF) *versus* spatial location for CR, ACR and FL regions displayed in **Figure 3** of the main text and T3 to T1 regions displayed in **Figure 5** of the main text. c-d. Map of incident photon to current efficiency *versus* spatial location for CR, ACR and FL regions displayed in **Figure 3** of the main text and T3 to T1 regions displayed in **Figure 5** of the main text. In both FF and IPCE values there are clear trends in the behaviour similar to the photocurrent density. The black/red region is the substrate and no calculation is made here.

Finally, we remark the incident photo to current efficiency (IPCE) is effectively a measure of a photoelectrodes external quantum efficiency [*J. Am. Chem. Soc.* 2006, 128, 49, 15714–15721]. For the best α -Fe₂O₃ electrodes *e.g.*, for ones prepared using atmospheric pressure chemical vapour deposition with passivation SnO₂ layers, IPCE values reach about 22% at 532 nm (and >1.4 V vs RHE) [*J. Am. Chem. Soc.* 2006, 128, 49, 15714–15721]. This shows that locally our defective electrodes do have morphologies present that can give rise to IPCEs are not only above that of the native film but also close to that of samples where the film preparation is optimised for performance. (A similar argument can be made somewhat for the fill factors [*Sol. Energ. Mat. Sol. Cell.* 27, 4, 1992, 335-345, *J. Phys. Chem. Lett.* 2012, 3, 9, 1208–1213, *J. Semicond.* 32 013001 2011]). In other morphologies (which dominate throughout the film) the FF/IPCE we obtain is low. The dominance of these low FF/IPCE regions means the overall PEC of the sample is quite low. **Our text has been modified as below to account for these changes:**

*‘Nonetheless, to ensure our local I-V curves link to performance we can use the local current density, local absorption and theoretical current extractable from the sample under solar illumination (see **Supplementary Note 3**) to obtain a semi-quantitative estimate of both the spatially varying fill-factor as well as the incident photon to current efficiency (IPCE)⁴⁹. The local fill factors vary between of 0.36 and 0.21, and the IPCEs from 22% to 10% ($\pm 4\%$), both in the order ACR>FL>CR, with a similar spatial distribution to the current density at 1.3 V_{RHE}*

(Supplementary Note 3). We can compare our local IPCEs to those previously obtained in the literature (at 532 nm and 1.5 V_{RHE}). Interestingly, in ACR regions our IPCEs reach close to that of high-performance α -Fe₂O₃ prepared by chemical vapour deposition⁴⁹. In other words, our defective film is in places comparable to an ‘optimised’ sample. While the trends in FF and IPCE underscore the link between the local I-V curves, water oxidation efficiency and morphology we emphasise that our experiments do not allow us to comment on the OER mechanism (see Supplementary Note 3 for further discussion). We note our trends reported above are also seen in other regions of the film near cracks (see Supplementary Note 4).’

The above discussion has been also added to the supporting information and is not reproduced for brevity. We direct the reviewer to pages 6-8 (Supplementary Note 3).

(3) The Raman maps in Figure 4d are somewhat challenging to interpret and could be improved by adding a contour overlay of the either the optical or SEM image (in order to better match the flat, crack, and adjacent to crack regions).

We thank the reviewer for this comment, contour maps have been generated from the optical images, with contour lines now faintly overlay on the Raman maps to improve clarity. These maps are shown in the new **Figure 4d** in our manuscript.

(4) The raw (not differential) Raman spectra associated with Figure 4c should be included in the SI, as smaller features in the differential spectra are difficult to see in the Figure. For example, the peak position of the 610 cm⁻¹ mode is shown in the lower left panel of Figure 4d, but this peak is not discernible in the differential spectra included in panel c. Additionally, the authors should clarify which Raman maps were collected under illumination.

We thank the Reviewer for highlighting this point. Examples of the raw spectra from characteristic regions of the sample have now been added to the supplementary information (new **Supplementary Note 8**) and are also detailed in the answer to question 4 part 1 of reviewer 2. The Raman spectra have also been analysed with and without light (under bias). Changes in key peak ratios are detailed showing some elements of structural rearrangements in the native films. This is also consistent with our μ -XANES experiments but require further study (see question 4.1 of Reviewer 2 for further discussion).

(5) Regarding the differential reflectance spectra (Figure 4a and 6b) the authors should assign the “two peaked structure” in the adjacent to crack and tier 2, tier 3, regions. For example, while the 550 nm feature could be a different type of oxygen vacancy (as discussed), what causes the redshift of the second feature from the assigned 570 nm towards 575-590 nm?

We agree with the reviewer that the two peaks observed in the differential reflectance spectra are not very common and interesting (also commented by reviewer 2). In general, we believe that assigning new absorption peaks unequivocally in any material is exceptionally challenging and out of the scope of the present study. However, we have performed an extensive series of experiments and literature search that have allowed us to suggest that such second peak might be related to another type of band-edge Fe *d-d* transition. Although we believe a completely new experimental and theoretical study would be required to confirm this. We refer the reviewer to comment 3 of reviewer 2 for our full discussion (this reviewer posed the same question and to improve legibility of the response we chose not to repeat the extensive discussion/new experiments provided there). With regards to the red shift specifically of the peak from 570 nm towards 575/580 nm in certain regions, we remark that this shift is first

relatively small. In the literature the reported band-edge of haematite varies between 2.1 and 2.2 eV (560 to 590 nm), depending on the exact film morphology. For example, extreme nanostructure in haematite films can blue-shift the band-edge [<https://arxiv.org/ftp/arxiv/papers/1111/1111.6204.pdf>], Sn⁽⁴⁺⁾ and C doping tends to red-shift the band edge [*ACS Omega* 2022, 7, 39, 35109–35117], specific facets will also have an influence on the band-edge, shifting it more than 0.2 eV [*RSC Adv.*, 2015, 5, 52210-52216] and other local features such as strain and electric field will also shift the band-edge. Given the limited energetic (2-4 nm) and spatial resolution (500 nm to 1 μ m) in many of our techniques, including the reflectivity measurements, we have decided not to interpret them. Instead, we focus more on interpreting the especially novel double-peaked behaviour. Nonetheless, in our revised text we have commented more on where the origins of this slight red-shift could arise from. **The following changes have been made:**

'In ACR there are two peaks, one at 550 nm and another at 575 nm (spectral resolution ~2 nm).

...

*The presence of the second differential reflectivity peak at 550 nm in ACR regions is challenging to assign. Comparison of our spectra to those in the literature, alongside electroabsorption and polarised reflectivity measurements (**Supplementary Note 6**), allows us to rule out its origin being from higher oxidation Fe species, Fe=O/Fe-OH/*O and specific nanostructuring⁵⁷⁻⁵⁹. Performing broadband fs-transient absorption spectroscopy on the same α -Fe₂O₃ samples (at open circuit potential; OCP) reveals that the decay kinetics are uniform in the spectral region associated with the peaks and that the additional peak could arise from a state that is present in the absence of bias (**Supplementary Note 6**). Combining this observation with literature calculations⁶⁰ and absorption measurements of epitaxial haematite⁶¹ suggests that the higher energy transition is related to a second type of hole species in the valence band of the semiconductor, i.e., another Fe d-d transition with similar electronic/chemical properties to the lower energy transition. However, further experimental and theoretical work is required for full assignment.'*

The above discussion is added to pages 11-15 of the supporting information (Supplementary Note 6).

(6) The authors suggest that the oxygen vacancy within the adjacent to crack regions could reside in some type of "intermediate sweet spot" balancing charge separation and negative trapping effects. Could the OV concentration here be determined by the Fe:O ratios identified in the EDX spectra?

We agree with the reviewer that being more quantitative with OV concentrations would be ideal. However, there are two challenges with using EDX to do this. Firstly, EDX (without a reference standard) is generally accepted to be a semi-quantitative method (but only under certain conditions). Quantification involves using the peak ratios to determine the relative proportion of components and then normalising them to 100% to account for all of a particular element. This method assumes the sample is flat (polished), homogeneous and 'infinitely' thick relative to the beam interaction volume. If these latter conditions are not met neighbouring components in the interaction volume will influence the results, i.e., there will be a topographic component. This will remain an issue to some extent even if external standards are used to calibrate the spectra. These so-called 'matrix corrections' are difficult in non-homogenous

samples like ours because of the need to find representative ‘background’ areas *e.g.*, of the same thickness, that are extremely homogeneous. Whilst other background subtraction methods exist, they tend to be less accurate. Furthermore, for quantitative EDX large accelerating voltages are often required to generate significant X-ray counts (up to 20 kV). We found these large voltages are particularly damaging to our samples. This is something critical for us to avoid as we are performing correlated microscopy measurements. Finally, even if all the above corrections can be performed the accuracy of EDX tends to be about 1%. Given the variation in oxygen vacancy concentration we expect is about 1-2% we believe we would be at the limit of our ability to detect the oxygen concentration accurately with EDX [<https://cfamm.ucr.edu/media/126/download?attachment>]. Indeed, it is for these reasons EDX is normally used for just compositional identification.

Whilst there will be a correlation between the O:Fe ratio from EDX and the oxygen content, EDX measurements will also be influenced by the atomic weight of the element trying to be detected. Light elements with atomic numbers (Z) <11 are particularly difficult to reliably quantify with EDX. Hence a direct comparison between EDX ratios and the oxygen vacancies measured by reflectivity measurements should be treated with caution. To calculate the concentration of oxygen vacancies, more complex techniques such as XPS could be used, however doing this with sub-10 μm resolution, as would be needed for us, remains challenging and well-beyond the scope of this work.

The following text, summarising the above discussion, has now been included in the manuscript to make the reader aware of the technical challenges/limitations associated with quantifying oxygen vacancy concentrations from small areas:

*‘However, we emphasise for such inhomogeneous samples, with light elements like O, the EDX values are only semi-quantitative and do not necessarily reflect only oxygen deficiency from vacancies (see **Supplementary Note 2** and below). Indeed, because of the sub-micron length scales we work at, quantitatively determining the local concentration of oxygen (or other elements) remains challenging, even by methods such X-ray photoelectron spectroscopy.’*

(7) The segregation of PCA1 and 5 under light illumination in Figure 4e is very interesting, however the lack of other data on this particular thin film makes it challenging to understand the origin of the spatially dependent features. The presentation of this data would be greatly improved if its chemical or potentiostat mapping were included here along with discussion of what could give rise to the spatial dependence.

We thank the reviewer for highlighting this point. Due to the technical challenges of performing *operando* μ -XANES it was near-impossible to perform correlated chemical or electrochemical mapping on exactly the same regions. However, to understand this result further we have performed a series of additional measurements.

The region examined by μ -XANES is effectively a homogeneous region of the α - Fe_2O_3 sample. What we observe from the XANES measurements is that under bias and light soaking there is a clear structural rearrangement of disorder, and potentially oxygen vacancies, in the films in this region.

To test this further, we have performed microscopic Raman imaging as a function of bias and light soaking. These experiments are detailed in **Figure R15-R18** and in response to question

4.1 of reviewer 2. However, for completeness we also repeat the salient points of the answer here.

The intensity ratio of the 293/441 cm^{-1} and 611/660 cm^{-1} Raman peaks has been previously found to correlate positively with the photocurrent density at 1.23 V vs RHE, the location of intraband trap states, band gap and flat band potential. In particular, the magnitude of I_{293}/I_{411} is suggested to correlate with the concentration of oxygen vacancies, whereas the magnitude of I_{611}/I_{660} has been linked to the level of structural disorder and population of intraband states [ACS Appl. Mater. Interfaces 2022, 14, 6615–6624, Chem. Sci., 2020, 11, 1085].

For the FL regions considered in **Figure 4** of the main text there is a systematic increase in I_{293}/I_{411} under illumination. For the I_{611}/I_{660} ratio there is a less clear change between dark and light conditions in FL but somewhat of a drop in the ratio, particularly as the potential is raised. No such change in either of these mode ratios is observed for the CR or ACR regions. It is challenging to fully understand this behaviour but given the link between the 293/441 cm^{-1} mode intensity ratio and oxide vacancies our results suggest in FL regions, under illumination and bias, there may be some changes in the vacancy population *i.e.*, vacancies are mobile. The mobility of oxygen vacancies in $\alpha\text{-Fe}_2\text{O}_3$ has not been measured to the best of our knowledge. However, in other oxide materials such as SrTiO_3 and TiO_2 the activation barrier height for oxide vacancy diffusion has been measured to be between 0.2 and 1.6 eV [Sci. Rep. 7, 46184 (2017), Phys Rev. Lett. 2012 109 (13)]. Interestingly, in these materials light is known to drive vacancy migration (when the material is held at bias) *via* Joule heating and the lowering of the heights of potential barriers associated with space charge zones depleted in charge carriers. Our observations suggest in ‘native’ thin films of haematite this may also be weakly possible, however further measurements *e.g.*, measurement of oxide vacancy barrier heights using STM or diffusion coefficients using impedance methods is needed. Finally, we note the microscopic lengthscales over which features like oxygen vacancies vary during the photoelectrochemical OER underscores the importance of moving of beyond bulk measurements, where such features are likely washed out and hence have been hitherto undetected.

We note that similar behaviour to that described above is observed for the T1 region which most closely resembles the native film (FL) *i.e.*, there is a change in I_{293}/I_{411} with/without light. (No such behaviour is seen for T2/T3). These results add further detail to our $\mu\text{-XANES}$ measurements as to the kinds of structural rearrangements that are occurring in regions of the native film under bias and illumination.

To understand if this behaviour can also be observed in the microscale electrochemical performance of samples, we have performed potentiostatic mapping after 5, 20, 60 and 120 min of continuous light soaking (400-900 nm) of the entire electrode under 1.5 V vs RHE bias (**Figure R4**). After 5, 20 min no clear patterns/segregation emerges in the maps of current density at 1.5 V vs RHE. However, after 60 mins some small local segregation in the photocurrent density emerges in the maps. After 120 min this appear to be still present but with little change. After ramping the films back to the open circuit potential, and leaving them in the dark for 10 mins, the maps of current density at 1.5 V vs RHE once again appear to be homogeneous. These results support the idea of some changes in structure/disorder from extended light exposure which can impact the overall local performance in the film. However, it is unclear exactly the mechanism for such reconstruction *i.e.*, is there underlying nanostructure in the film that enables this? Answering this is out of the scope of the present work.

Figure R4: Maps of photocurrent at 1.5 V vs RHE across homogeneous region of hematite film shown in **Figure R6**. Each pixel is 1 μm . The time indicates the period for which the electrode is illuminated with a Xe white light $500 \mu\text{m}^2$ illumination area.

Figure R5: $t_{50\%}$ values at 1.6 V vs RHE across a selection of pixels indicated with red stars in **Figure R6**. Error bars derived from standard deviation across values.

Figure R6: Brightfield optical image of haematite thin-film. Scale bar is 1 μm . Red asterisks show locations where transient kinetics are measured.

Finally, we did also measure the $t_{50\%}$ values and reflection spectra after 5, 20, 60 and 120 min of continuous light-soaking (400-900 nm light as above over the entire electrode area) at 1.5 V vs RHE (**Figure R5**). However, we found no statistically significant changes in these measurements over time. This may be related the fact that the pathways are not necessarily affecting the recombination dynamics or the reduced sensitivity of the transient reflection microscopy as compared to XANES. Indeed, given the uncertainty on $t_{50\%}$ values is $\sim 10\%$, which is similar to the spatial variation in photocurrents, this may be the main reason why we cannot observe any differences. Regardless, further experiments particularly with regards to quantifying oxygen vacancy mobility in haematite, are necessary.

With all the above in mind it is important to consider the illumination time during a measurement in our analysis. The μ -XANES measurements involve illumination of the whole area of the electrode whilst scanning the X-ray beam and take approximately 60 mins per map. Hence, it is understandable that in these measurements the effects of light induced oxide migration can be present. The transient reflection microscopy measurements involve point scanning *i.e.*, there is only light on the sample for the \sim 30 s it takes to collect of a transient kinetic at a given spot. Hence, light-induced structure rearrangement effects cannot influence the results of this measurement.

The reflectivity measurements require <5 mins collection/illumination time across the whole water oxidation range and hence cannot be influenced by light induced structure rearrangements. The Raman imaging we perform (without light, but with bias) involves off-resonant excitation which will not contribute to the structural rearrangement (and involves point scanning with <10 s collection time per spot). The light and biased based Raman imaging does involve continuous soaking of the entire electrode for 50 mins before starting of measurements with a <60 min entire measurement time (including ramping across biases and varying measurement locations). It is hence feasible in these measurements that light induced structure rearrangements can be observed. Photoelectrochemical mapping involves point scanning with 5-10 min of illumination per sample location (across an entire sweep of biases), hence unless explicitly tested for, as above, this measurement should not be influenced by any light induced oxide vacancy migration. In the case of photoelectrochemical mapping with light-soaking/external illumination, maps were only measured at 1.5 V_{RHE} for rapid acquisition (<5 min full map acquisition).

The timescales for measurements have been added to the **Methods** section of the manuscript and it is now stated that results are not influenced by time period of illumination during a measurement unless this effect is being explicitly tested for.

The following changes to the main text have been made to account for the above new data:

'Repeating the above measurements under continuous illumination (400-550 nm; electrode pre-light soaked for 50 mins) shows limited difference in the response except in the FL regions (Supplementary Note 8). Intriguingly, a clear increase in the intensity ratio between E_g modes at 293 cm^{-1} and 411 cm^{-1} , linked to the degree of disorder and oxygen vacancies in $\alpha\text{-Fe}_2\text{O}_3$, is observed here^{66,67}. This tentatively suggests that light can accentuate structural (disorder) and vacancy rearrangement in 'homogeneous' regions of the sample under bias (see Supplementary Note 8 for further discussion).

...

As discussed, these features could be linked to the OV population, suggesting that OVs are dynamically re-distributing under light and bias. This observation is also reinforced by measurement of photocurrent maps of FL-like (homogenous) regions over extended time periods (Figure 4f). Following >60 mins of continuous white light exposure under bias (400-900 nm over the whole measurement area) we find a slight segregation of photocurrent values which again suggests structural rearrangements might be occurring (see also Supplementary Note 8). We note that as the measurement/illumination time in all other measurements detailed in this work is significantly shorter than the timescale over which we observe changes induced

*by bias and light, they will not be influenced by this effect unless stated (see **Methods and Supplementary Note 8** for further details).*

...

The light-driven migration/segregation of disorder (potentially OVs) we observe is intriguing but characterising the (local) oxide vacancy diffusion coefficient and activation barrier will be instrumental in learning how to exploit this phenomenon.'

In addition, Figure 4 of the main text has additionally been altered to include photoelectrochemical maps as a function of external light soaking time. We believe this data provides useful and interesting complements to the μ -XANES data, and further strengthens our argument about light based structural rearrangements.

Additional information has now been added to pages 24-24 of the supporting information (Supplementary Note 9).

(8) The authors should check the scale bar for the Ag1 mode in Figure 6c, as it seems odd that the peak frequency would extend all the way to 0 cm^{-1} .

We thank the reviewer for noting this confusing scaling. The Raman spectra on parts of the substrate where there is no sample were previously set to zero. There is no material in this region and hence fitting peak frequencies, intensities or widths is not appropriate. This region is no longer included in the colorscale/range to minimise confusion and Figure 6c of the main has now been corrected.

Reviewer #2 (Remarks to the Author):

Authors developed in-situ spatial imaging methodology (photocurrent, spectroelectrochemical reflection spectroscopy, transient reflection spectroscopy, Raman spectroscopy, and X-Ray near-edge absorption microspectroscopy) to reveal the spatial heterogeneity of reactivity and establish robust structure-function relationships. Using prototypical Fe₂O₃ photoanode, the correlation between oxygen vacancy concentration (crack and thickness) and photoelectrochemical performance is established. This work is novel and provides some insights for understanding activity and the impact of defects from microscopic mapping, which could be recommended for publication in Nature Communication after addressing the following comments.

We thank the reviewer for highlighting our work as being novel and that we provide practical insights as well as a step forward in multi-modal analysis of photoelectrodes. We thank the reviewer for recommending publication in Nature Communications after addressing their comments, which we have done extensively below.

Comment #1

Fe₂O₃ film was prepared by solvothermal (FeOOH) and following annealing (500 °C, Fe₂O₃). (1) This film wasn't activated by high temperature, and the high PEC performance is puzzling. A macroscopic PEC performance needs to be provided. (2) According to previous reports and Supplementary Note 1, this film usually shows irregular nanorod morphology and isn't dense film. The discussion about the impact of thickness needs to be careful.

The reviewer is indeed right in pointing out we did not perform any further high temperature treatment. This was deliberate as our goal was to create a rather defective hematite film with a significant number of oxygen vacancies allowing us to study them under the different morphological conditions discussed in our manuscript.

Performing macroscopic I-V curve measurements (cm diameter beam spot, 1 sun white light illumination) across several films/locations we find current densities at 1.6 V vs RHE of between 0.9 and 1.1 mA cm⁻² depending on the location, with an average value of 0.97 mA cm⁻² (see **Figure R7a** for non-local/macroscopic I-V performance curve). Based on the work of Zandi and Hamann [*J. Phys. Chem. Lett.* 2014, 5, 9, 1522–1526] we might expect the non-locally measured current density at 1.5 V_{RHE} to be lower at ~0.6 mA cm⁻² for such a low temperature (500 °C)-annealed film. For a higher temperature (800 °C)-annealed sample it should be around ~0.8 mA cm⁻² based on this reference. However, these values vary in the literature with Mesa *et al.*, reporting much higher current densities for such samples around 1.1 mA cm⁻² [*J. Phys. Chem. Lett.* 2020, 11, 17, 7285–7290]. This suggests that our (macroscopic) I-V response is not necessarily unusual compared to the literature, but that the films we examine may not be as defective as a traditional, solvothermal preparation-based, defect rich, films.

Moreover, the fact that the local I-V response (0.7 to 1.3 mA cm⁻² at 1.5/1.6 V vs RHE) lies above and below the range of the microscopic one, further supports the claim that there are regions of the film that locally can perform as well as ones that were annealed, *i.e.*, not all microscopic defects are detrimental to the performance.

We also note that our onset potentials are more in-line with low-temperature annealed films at between 0.8 and 1.1 V vs RHE compared to the 0.6/0.7 V vs RHE that would be expected for high-temperature annealed films. It is additionally important to bear in mind that as we are

comparing relative current density/onset potentials between film locations, as opposed to absolute values, any discrepancies do not actually affect the conclusions of our work. Generally, we need to be careful when comparing our results between other studies due to the different ways in which macroscopic PECs can be measured. For example, in some cases films can be masked which will influence the overall performance, especially when there are large inhomogeneities as is the case of defective films. This will then change the reproducibility of values.

We have also performed new measurements to obtain local incident photon to current efficiencies (IPCE) as well as fill factors (based on the current densities achievable theoretically from our samples; see response to question 2 of reviewer 1). We find values of between 10% and 22% for the IPCE in-line with a defect rich film. We have made note of all the above in our revised text, and in-particular the slightly larger than expected PEC performance. **The following modifications have been made:**

*‘Interestingly, the current densities in **Figure 3b** are slightly higher than might be expected based on other studies of hydrothermally synthesised $\alpha\text{-Fe}_2\text{O}_3$ e.g., 1.2 mA cm^{-2} at $1.5 V_{\text{RHE}}$ here vs 0.7 to 1.1 mA cm^{-2} at $1.5 V_{\text{RHE}}$ in ref⁴⁸. But the ‘local’ current densities do match with macroscopic measurements of the sample (**Supplementary Note 2**), and have OER onset potentials that would be expected for such defective films⁴¹.’*

We also wish to note that there was slight miscalculation in our original submission of the photocurrent maps for T1, T2 and T3. Furthermore, in our original submission we did not account for the differing degrees of absorption for the various layers when normalising our photocurrent response (see detailed response to question 3 of reviewer 3). Consequently, while our photocurrent response does remain high, it is below 1.3 mA cm^{-2} . We apologise for the confusion this error may have caused and all values have been carefully checked now.

Figure R7: **a.** Macroscopic I-V curve of typical haematite electrode measured in this work at 1 sun illumination. **b.** Photocurrent at 1.6 V vs RHE of identically prepared (low-temperature annealed) haematite electrodes.

We further agree with the reviewer that caution needs to be exercised when relating the morphology of films to activity. This is because there may be nanoscale morphology which we cannot resolve that may also be playing a role. We note that the nanorod morphologies present in non-annealed haematite films are only clearly visible at the sub-100 nm lengthscale (~50 nm wide nanorods; see more detailed SEM images in Figure S2 of *J. Phys. Chem. Lett.* 2020, 11,

17, 7285–7290). As in our work we are focussing on *microscale* (1 μm and above) structure that is accessible with optical and X-ray techniques we cannot comment on the impact of such features. Whilst we agree the low-resolution SEM images in **Supplementary Figure 1a** do not show a nanorod morphology, some new higher resolution images of the films are now included. These images are still only at approximately 500 nm resolution level (**Figure R8**) but show the emergence of nanorod-like features on the surface as would be expected based on *e.g.*, *J. Phys. Chem. Lett.* 2020, 11, 17, 7285–7290. We note that even under the same preparation conditions subtle variations in the haematite performance/properties can emerge. Hence, we do not believe it is surprising that our film does not behave exactly as all previous reports (see variation in photocurrent between *J. Phys. Chem. Lett.* 2014, 5, 9, 1522–1526 and *J. Phys. Chem. Lett.* 2020, 11, 17, 7285–7290., where the same synthetic route is reported but the maximum photocurrent at 1.6 V vs RHE varies by almost 0.5 mA cm⁻²).

Figure R8: SEM image of an $\alpha\text{-Fe}_2\text{O}_3$ photoelectrode studied in this work showing sub-micron nanorod features emerging on the surface.

To acknowledge that the distribution of these nanorods may also play a role in the observed behaviour, particularly when assessing the effect of thickness, we have reworded our manuscript. We agree with the reviewer that we need to be careful such that our results on thickness effects are not influenced by simple absorption differences in the different regions of the sample. To this end we have reanalysed our data and performed new measurements as detailed in the response to question 3 of reviewer 3.

The main text has been modified to make note of the above as follows:

*‘We also note that there will be some sub-300 nm morphology present in our films (*e.g.*, nanorods see **Supplementary Note 2**) which we cannot explore, due to the spatial resolution of the methods explored here.’*

Pages 4-5 of the supporting information (Supplementary Note 2) now include the above discussion also.

Comment #2

Oxygen vacancy usually shows two functions, charge recombination and surface reaction. In manuscript, the authors only considered the impact of charge recombination, and ignored the effect on surface reaction kinetics. (1) In Figure 3, FL, ACR and CR show similar photocurrents at 1.5 VRHE, but the half lifetime ($t_{50\%}$) and amplitude from CR are obviously different from

those of FL and ACR. These differences can't be understood by charge recombination. (2) In line 150-155, authors correlated the onset potential of photocurrent with back electron transfer reactions. This correlation is inaccurate, because severe recombination in bulk Fe₂O₃ photoanode can lead to large onset potential. The transient photocurrents and integral charges should be provided to compare back charge concentration in CR, ACR and FL.

We thank the reviewer for pointing this interesting question out. Indeed, several publications relate oxygen vacancies with surface states that participate in the OER mechanism [*J. Am. Chem. Soc.* 2012, 134, 9, 4294–4302, *ACS Appl. Mater. Interfaces* 2015, 7, 31, 16999–17007, *Angew. Chem.* 58, 2019]. However, more recent kinetic analysis that involves correlating the measured optical absorption of photogenerated holes with the measured rate of OER, suggests that oxygen vacancies seem to have a more significant impact in recombination than in the surface reaction process (see *J. Phys. Chem. Lett.* 2020, 11, 17, 7285–7290). In this paper, we studied the rate determining step (RDS) of the OER in 4 different hematite photoanodes, with different optical and electronic properties (including doping density and oxygen vacancies). Kinetic analyses revealed a 3rd order process with respect to surface hole density for all the samples studied, suggesting that the RDS is achieved when 3 photogenerated holes are accumulated at the reaction centre (see more details in *Nat. Chem.* 12 82-89 2020). Additionally, in these studies, experiments at different applied potentials (*i.e.*, different filling states of the oxygen vacancies) revealed the RDS follows also a 3rd order of reaction, which suggests that the reaction mechanism does not change, regardless the amount of oxygen vacancies or their filling state. As such, we believe that although oxygen vacancies can participate in the surface reaction, as coordination sites or to accumulate surface holes, their role is more important in recombination losses.

Whilst we agree that a local transient photocurrent (TP) mapping experiment would be interesting to perform on these samples, to further investigate the role of surface kinetics, this is highly challenging to do accurately with microscopic resolution. Indeed, to the best of our knowledge very few studies exist that do this and none have been performed on photocatalytic systems. The main problem is that analysis of TP measurements relies on the excitation being a small perturbation [*J. Phys. Chem. C* 2012, 116, 51, 26707–26720]. Whilst this is true when performing macroscopic measurements, in the case of local focussed excitation, a relatively large excitation density is required to obtain a measurable signal. The interpretation used for macroscopic signals is no longer valid and analysing the data requires a new framework to be built (which would be a separate study in itself). We note that for the transient reflection microscopy we perform the issue of high excitation densities is less of an issue for interpretation of the kinetics (especially as we stay at the limit of the linear excitation regime; see **Supplementary Note 14**). Furthermore, empirically (at the ensemble level) this all-optical measurement is easier to obtain a signal from, at low excitation densities, than TP. Although, as we acknowledge in the conclusion of our work, ideally this measurement would be performed at lower excitation densities and with an LED to better capture solar conditions [*J. Chem. Phys.* 153, 150901 (2020)]

Another point that should be noted is that in the few cases when transient photocurrent mapping has been performed it has been on relatively homogeneous samples (2D materials, photovoltaic thin-film blends) [*e.g.*, *Sci. Adv.* 2022, 8, 50]. This is because optical scattering can influence the measured transients. This would be a further cause for concern when trying to perform such measurements on our samples [*Sol. Energy Mater. Sol. Cells*, 161, 2017, 89-95]. Consequently, we believe such measurements would not necessarily provide reliable data for the microscale regions we are analysing. The development of this technique (and associated analysis) could

be the basis of a future study. Indeed, in this work we already develop a relatively brand-new transient reflection microscopy method for analysing material dynamics (thus far transient reflection microscopies have focussed on the fs to ns regime using an inherently different detection and readout to that presented here).

Nonetheless, we fully acknowledge and agree with the reviewer's point that there is some uncertainty with regards to the full interpretation of our $t_{50\%}$ values, and the role played by surface kinetics. Further work is needed to establish this, and generally the full origin of the local enhancements in performance we observe. This will arguably involve the development of microscopic TP mapping. However, regardless of the exact origins of the behaviour, the principal conclusion of our work that there is a strong spatial dependence in the performance of haematite films at the micrometre scale, is still valid. Furthermore, our paper highlights the point that one needs to understand the microstructure to really understand the role of oxygen vacancies in thin-film photoelectrocatalysis. Hence, even studies at the macroscale where for example oxygen vacancies are controllably introduced during synthesis and the surface recombination is then probed with TP, might not be sufficient. Further methodological work is required such that local TP measurements can be performed and interpreted.

To make clear that there is some uncertainty in the interpretation of our results we have revised our discussion of the data presented in Figure 3 and made note now that oxygen vacancies could also be modulating the surface kinetics and hence the $t_{50\%}$ values we obtain.

The changes to the main text are as follows:

*'Finally, we note that OVs can also influence surface reaction kinetics, acting as coordination sites or sites to accumulate surface holes at. However, recent studies, on hematite in a range of morphologies, have revealed that the reaction mechanism does not change, regardless of the filling/density of OVs⁴¹. This implies that there is a more dominant role of OVs in recombination losses rather than on the surface reaction kinetics. Nonetheless, surface reactions linked to OVs may play some role in the hole dynamics detailed in **Figure 4c**, and further work using e.g., microscale transient photocurrent mapping is required to fully establish the extent of this. We however note further methodological developments in terms of working at low excitation densities and in rough morphologies⁶² are required before such measurements can reliably be performed at the microscale (see **Supplementary Note 3**).'*

The above discussion has been added to page 8 of the supporting information (Supplementary Note 3).

Comment #3

In lines 184-188 and in Figure 4a, the peak assignment of 550 nm to a second type of hole species at dark is arbitrary. Because Figure 4a suggests that the lower potentials can induce the appearance of this peak, lower potentials can't generate high-energy holes. This peak need to be re-assigned.

We agree with the reviewer that assigning this peak would be of great interest (as also commented on by the other reviewers). However, generally assigning new absorption peaks unequivocally in any material is exceptionally challenging, often involving consensus at the community level, both experimentally and theoretically. Hence, here we have focussed on performing new experiments that might provide some insight on the species responsible and ruling out other sources:

Defect/trap site absorption: We have performed macroscopic potential dependent absorption measurements on ultrathin (20 nm) hematite films that are qualitatively highly defective. The films are hydrothermally synthesised and annealed at 500 degrees which alongside their thinness results in the high defect concentration. In this case we also observe two peaks in the absorption spectra. However, (i) the peaks are not present strongly until 0.9 V vs RHE (see **Figure R9**), in contrast to what is observed in the current manuscript where the peaks are observed at biases lower than 0.6 vs RHE and (ii) the peaks are separated by ~80 nm as opposed to ~30 nm in the current text. We consequently suggest the two absorption peaks likely do not come from high-energy mid-gap defect/trap/vacancy site absorptions that are created by a bias.

We note a double peaked absorption spectrum, where the peaks are separated by 120 nm (460 nm and 580 nm peaks) and grow in with bias, has been observed by Klahr and Hamann [*J. Phys. Chem. C* 2014, 118, 19]. In this case the response was linked to the absorption of Fe=O and Fe-OH surface states. However, given this spectrum is quite distinct to the one we observe, we can rule out this effect.

The above noted paper did also report a peak splitting of the central absorption peak at 580 nm into two peaks at 560 ± 5 and 570 ± 5 nm. This is narrower than we observe and also was only seen to occur in certain potential ranges. Calculations by Snir and Toroker [*J. Chem. Theory Comput.* 2020, 16, 8, 4857–4864] demonstrated that these peaks arise from *OH and *O surface species which we can also rule-out.

Figure R9: Macroscopic potential dependent absorption measurements of highly defective 20 nm thick haematite films.

High oxidation state Fe species: Fe(IV), Fe(V), and Fe(VI) species show the following absorptions in solution in the 400-700 nm spectral region: Fe(IV) shows a broad absorption peak at 420 nm, Fe(VI) has a broad peak at 510 nm, Fe(V) has a main peak at 400 nm and a second peak at 500 nm [*Science* 315, 5813, 2007 835-883, *J. Am. Chem. Soc.* 2012, 134, 2, 1228–1234]. None of these peaks match with our observations and hence we can likely rule out their role in giving rise to the double-peaked spectrum we observe.

Kinetics of two absorption peaks: We have performed spectrally resolved transient absorption measurements (without bias) on several locations of the haematite films. Interestingly in certain regions of the film a double peaked feature appears in the transient absorption spectra (355 nm pump; 200 nJ; 200 fs time resolution), with the peak centres closely matched with those of the steady-state reflectivity as a function of voltage ($\Delta R(V)$) detailed in the main text. The transient absorption spectra report mainly on the photoinduced absorption (PIA) bands of $\alpha\text{-Fe}_2\text{O}_3$, whereas in the $\Delta R(V)$ spectra we are examining ground state absorptions. The two are hence not directly comparable. Indeed, because the ground state bleach of hematite is particularly short (<1 ps) it becomes obscured by the PIA. However, there will be an underlying ground state bleach kinetic within the transient absorption spectra which will reflect the lifetime of the two states we are sensitive to in the $\Delta R(V)$ spectra. Furthermore, the structure we observe in the PIA bands can also be created by an underlying absorption structure that is reflected in the bleach, *i.e.*, is not from two distinct PIAs. Altogether the above makes it challenging to directly assign the transitions we are observing in the transient spectra and correlate them with the data we see in the $\Delta R(V)$ spectra. But: (i) the uniform decay kinetics across 500-700 nm wavelengths suggest the states responsible for the two peaks have very similar lifetimes and (ii) may be related to some underlying states present in certain regions in the ground state of the material (*i.e.*, present without bias).

We note that performing spectrally resolved transient reflection mapping remains technically challenging so we cannot correlate the spectra in **Figure R10** with exact regions of the film. However, to further verify point (i) in the above we have repeated transient reflection measurements in the T3 region detailed in **Figure 5** of the text at 540 nm and 590 nm (at 1.6 V vs RHE). Identical decay kinetics are found to those previously detailed at both wavelengths (**Figure R11**).

Figure R10: a-b. Spectrally resolved transient absorption spectra of haematite thin films measured in this manuscript at locations showing single peaked photoinduced absorption bands (a) and double peaked bands (b). c. Kinetics at 545 nm and 575 nm from transient absorption spectra in b.

Figure R11: Transient reflection kinetics of ACR region detailed in **Figure 5** and **6** of main text, measured at 540 nm 590 nm and 638 nm.

Polarizability and dipole moment

We have performed electroabsorption (EA) of measurements on the haematite films to understand how the band-edge transitions are sensitive, if at all, to electric fields or if they can be revealed more clearly by the presence of a field. To create devices for electroabsorption spectroscopy the FTO glass was coated with 70 nm of Al_2O_3 , haematite was deposited as in the main text, 70 nm of Al_2O_3 was then placed atop of the film with a 15 nm Cr/Au semi-transparent contact pad. Electroabsorption measurements were performed in an ensemble transmission geometry using the same setup as previously detailed in [*Nat. Commun.* 11, 5617 (2020)]. All measurements were performed at a fixed field of 600 kV/cm as this produced sufficient signal-to-noise ratios. Most simply the EA signals can be modelled as a combination of linear:

$$\Delta\alpha = -\frac{1}{2}\Delta p F^2 \frac{d\alpha}{dE}$$

where $\Delta\alpha$ is the EA signal, $\frac{d\alpha}{dE}$ the first derivative of the absorption spectrum, Δp the change in polarizability and F the electric field, and non-linear terms:

$$\Delta\alpha = -\frac{1}{6}\Delta\mu^2 F^2 \frac{d^2\alpha}{dE^2}$$

where $\Delta\mu$ is the change in dipole moment and $\frac{d^2\alpha}{dE^2}$ the second derivative of the absorption spectrum. While more sophisticated models exist to analyse the electroabsorption data, for our exploratory measurements we stick to this well-applied model [see *e.g.*, *Ann. Rev. Phys. Chem.* 48 213-242 1997]. Applying this analysis model to the electroabsorption spectra shown in **Figure R12** we find relatively good agreement between the EA and the weighted derivative spectra. In other words, any additional band-edge transitions do not become more clearly visible in EA. This may be due to their insensitivity to the field, weak oscillator strength/forbidden selection rules or limited sensitivity of the technique itself. Nonetheless the EA spectra do allow us to estimate that in $\alpha\text{-Fe}_2\text{O}_3$ the Fe d - d band-edge excitons at 590 nm have a $\Delta\mu$ of 2.4 D and Δp of 1.5 \AA^3 (35% fitting errors on both). This is in-line with rather localised electron-hole pairs with a charge-transfer like excitonic-character (from the large dipole moment) as would be expected for a material like $\alpha\text{-Fe}_2\text{O}_3$. Finally, we remark that

further analysis is required to fully characterise electric field effects of transitions in the material *e.g.*, field-dependent measurements and new modelling, but these are beyond the scope of the current work.

Figure R12: Electroabsorption spectrum of α -Fe₂O₃ (solid blue line) with associated first and second derivative fit (orange dotted line). **b.** Corresponding film absorption.

Theoretical basis for double peaked absorption: Several theoretical works examine the absorption spectrum of hematite using *GW BSE* electronic structure calculations [*Phys. Chem. Chem. Phys.*, 2019, 21, 2957]. Interestingly, all works report the presence of two closely separated optical transitions, separated by 0.15 eV around the band-edge. These transitions are sometimes assigned to originating from the same or different states (direct and indirect transitions) and have a similar spacing to that observed in our work. Whilst the evolution of these transitions with bias has not been examined, they have experimentally been observed in the absorption spectrum of films with nanorod morphology, as we have [*J. Electrochem. Soc.*, 147 (7) 2456-2461 (2000)]. We consequently can tentatively suggest that the two peaks we observe may be related to different hole transitions. We note given we observe similar decay times in the transient absorption spectra across the spectral range that is likely associated with these hole transitions, it is likely they originate from the same electronic state, *i.e.*, are not necessarily distinct direct and indirect transitions.

To shed further light on where these transitions may arise from we can turn to the work of Marusak *et al.* [*J. Phys. Chem. Sol.* 41, 9, 1980, 981-984]. Here, they measured the absorption spectra of α -Fe₂O₃ (110 crystal face) with high-sensitivity at zero bias and analysed the spectra using ligand-field theory. They also found two closely spaced transitions (albeit separated by 0.22 eV) and ascribed them to transitions between ⁶A₁ -> ⁴A₁ (high-energy transition) and ⁶A₁ -> ⁴E (lower energy transition) symmetry states in the ligand field of the iron atoms, *i.e.*, Fe (3d) valence band to Fe (3d) conduction band transitions. The energy separation of the transitions we observe matches well with both the above description and our spectra. The difference in intensity/presence between peaks in the voltage differential reflectivity spectra may then be a consequence of the selection rules and strength/presence of these transitions in a particular region of the film. In any case the work of Marusak *et al.* and [*Phys. Chem. Chem. Phys.*, 2019, 21, 2957] suggests that we are observing transitions between two states of the

crystal field of the iron centres in hematite, *i.e.*, two different hole species. These species appear to involve both similar electronic states and have a similar chemical behaviour, *i.e.*, we can tentatively suggest they show the same general OER mechanism.

It is unclear what drives the presence of this double peaked spectrum in our films. For example, our T1 region which we suggest is rich in oxygen vacancies does not show this peak, whereas the similarly oxygen vacancy rich ACR region does. This suggests this behaviour is not linked only to the oxygen vacancy concentration/presence. Similarly, structurally there does not seem to be a consistent signature for this behaviour response *e.g.*, the A_g/E_g mode intensity is high in T2/T3 but low in ACR. What is common to the appearance of this behaviour is high $t_{50\%}$ values, suggesting it is related to the underlying electronic states involved/accessible in the recombination/after photoexcitation. However, we believe commenting on this further is beyond the scope of this work.

Impurity phases: Other phases of hematite *e.g.*, β -Fe₂O₃ and ϵ -Fe₂O₃ have band-edges at 1.9 eV and 1.35 eV, respectively. These are well away from that of the samples we examine and do not appear to contribute [IOP Conf. Ser.: Mater. Sci. Eng. 881 012099 (2020)].

Faceting/nanostructuring: To rule out effects from nanostructuring/preferential faceting we have performed polarised absorption of films in ACR-like regions at 0.8 V vs RHE examining the intensity of the two peaks as a function. (Fixed polarisation incoming light, varying polarisation in detection). We find no dependence on the detection polarisation, suggesting there is no preferential alignment in the microstructure of the films.

Figure R13: Magnitude of change in reflectivity (maximum normalised intensity) at 0.8 V vs RHE of high and low-energy peaks of double peaked reflectivity spectra in ACR regions of sample. Reflectivity change is normalised to ΔR at a 0 degrees polarisation angle for low energy peak. Error bars are not shown but are approximately 15% of the values based on the signal-to-noise-ratios in the experiment.

In summary whilst we cannot unequivocally assign the second high-energy peak, our extensive new experiments and literature survey suggest it arises from a state related to band-edge Fe *d-d* transitions, much like the well-established low energy peak at 2.1 eV. Further calculations, experiments and community consensus, beyond the scope of this work, will be required to fully assign this additional transition. Generally, we remark that while for a metallic electrode the energetics may depend on the bias, in the defective oxide electrodes we are working with, the morphology, as we have shown, can also play a large role, *i.e.*, bias does not necessarily solely determine energetic filling/ordering.

The following changes have been made to the main text:

*'The presence of the second differential reflectivity peak at 550 nm in ACR regions is challenging to assign. Comparison of our spectra to those in the literature, alongside electroabsorption and polarised reflectivity measurements (Supplementary Note 6), allows us to rule out its origin being from higher oxidation Fe species, Fe=O/Fe-OH/*O and specific nanostructuring⁵⁷⁻⁵⁹. Performing broadband fs-transient absorption spectroscopy on the same α -Fe₂O₃ samples (at open circuit potential; OCP) reveals that the decay kinetics are uniform in the spectral region associated with the peaks and that the additional peak could arise from a state that is present in the absence of bias (Supplementary Note 6). Combining this observation with literature calculations⁶⁰ and absorption measurements of epitaxial haematite⁶¹ suggests that the higher energy transition is related to a second type of hole species in the valence band of the semiconductor, i.e., another Fe d-d transition with similar electronic/chemical properties to the lower energy transition. However, further experimental and theoretical work is required for full assignment.'*

The supporting information has been changed on pages 11-15 (Supplementary Note 6).

Comment #4

(1) Imaging and spectra of Raman are only measured under OCP and potential-dependent conditions. In order to understand the impact of oxygen vacancy on PEC performance, the imaging and spectra of Raman under illumination-dependent conditions need to be provided.

We thank the reviewer for highlighting this. **The raw Raman spectra are provided below and have been added to the supporting information on pages 18-22 (Supplementary 8).** We have performed new Raman measurements (off-resonant excitation 785 nm) under illumination. The illumination is from a white light with a 550 nm short-pass filter to prevent the illumination light interfering with the Raman signal (400-550 nm illumination). The whole area of electrode is illuminated simultaneously, and electrodes are pre-soaked with light for 50 mins before illumination to match the timescales of XANES measurements.

For the ACR/CR regions considered in **Figure 4** of the main text there is limited evolution of the Raman spectra under light. However, interestingly for the FL regions the intensity ratio of the 293/441 cm⁻¹ and 611/660 cm⁻¹ Raman peaks is (reversibly) altered. The ratio of these particular modes has been previously found to correlate positively with the photocurrent density at 1.23 V vs RHE, the location of intraband trap states, band gap and flat band potential. In particular, the magnitude of I₂₉₃/I₄₁₁ is suggested to correlate with the properties of oxygen vacancies, whereas the magnitude of I₆₁₁/I₆₆₀ the level of structural disorder and population of intraband states [ACS Appl. Mater. Interfaces 2022, 14, 6615–6624, Chem. Sci., 2020, 11, 1085].

For the CR and ACR regions there is limited change in I₂₉₃/I₄₁₁ and I₆₁₁/I₆₆₀ in the light and dark (**Figure R15**). However, interestingly for the FL region there is a systematic increase in I₂₉₃/I₄₁₁ under illumination. For the I₆₁₁/I₆₆₀ ratio there is a less clear change between dark and light conditions in FL but somewhat of a drop in the ratio particularly as the potential is raised. It is challenging to fully understand this behaviour but given the link between the 293/441 cm⁻¹ mode ratio and oxide vacancies our results suggest in FL regions under illumination and bias there may be some changes in the vacancy population i.e., the vacancies and structural disorder are mobile. The mobility of OVs/disorder in α -Fe₂O₃ has not been measured to the best of our

knowledge. However, in other oxide materials such as SrTiO₃ and TiO₂ the activation barrier height for oxide vacancy diffusion has been measured to be between 0.2 and 1.6 eV [*Sci. Rep.* 7, 46184 (2017), *Phys Rev. Lett.* 2012 109 (13)]. Interestingly, in these materials light is known to drive vacancy migration (when the material is held at bias) *via* Joule heating and the lowering of potential barriers associated with space charge zones that are depleted in charge carriers. Our observations suggest in ‘native’ thin films of haematite this may also be weakly possible, however further measurements *e.g.*, the measurement of oxide vacancy barrier heights using STM or diffusion coefficients, using impedance methods, is needed. Finally, we note the microscopic lengthscales over which features like oxygen vacancies/disorder vary during the photoelectrochemical OER underscores the importance of moving of beyond bulk measurements, where such features are likely washed out and hence have been hitherto undetected. We also emphasise that given the limited quantification provided by Raman on features like OVs and general disorder, we cannot be sure that it is exactly OVs that are changing under light soaking or whether the changes we observe arise from other material properties (see answer to question 4.2 of the reviewer).

For the T1/T2/T3 regions systematic changes in Raman peak ratios are harder to detect (**Figure R17**). For I₂₉₃/I₄₁₁ in T2/T3 there is very little change with and without light. Whereas for the T1 region which most closely resembles the native film (FL), there does indeed appear to be a change in I₂₉₃/I₄₁₁ with/without light. For the I₆₁₁/I₆₆₀ the changes with illumination are less clear, however this intensity ratio drops both in T3 and somewhat in T1 under light soaking and bias. These results broadly support the hypothesis that there is structural rearrangement in regions of the native film under bias and illumination. The exact nature of all of these remains to be further investigated.

We note that the above results are consistent with our μ -XANES measurements in **Figure 4** where structural changes were observed in the native film under bias and illumination. However, as we state throughout our work care has to be taken when comparing across techniques. For example, although the FL/T3 regions tend to show the greatest differences in our bias dependent Raman when illumination is added, they show a relatively small change in bias dependent reflectivity. However, this latter measurement is effectively done in the dark, *i.e.*, it is not measuring the effect of illumination on the OVs simply their population and how it responds to bias.

As a final point we wish to note that we have several different conditions under which Raman imaging is performed. Raman imaging without light, but with bias involves off-resonant excitation which will not contribute to the structural rearrangement (and involves point scanning with <10 s collection time per spot). The light and biased based Raman imaging does involve continuous soaking of the entire electrode for 50 mins before starting of measurements with a sub 60 min entire measurement time (including ramping across biases and varying measurement locations). These conditions mimic closely the μ -XANES, potentially explaining why similar features are detected between the two.

Figure R14: a-c. Raw Raman spectra from FL (a), CR (b) and ACR (c) regions described in the main text, under bias dark/light (sub-550 nm wavelength of illumination over an area greater than $500 \mu\text{m}^2$). For the CR region the spectra have been averaged over 4 pixels to improve the signal-to-noise ratio and in the ACR region over 9 pixels. The spectra are truncated at 800 cm^{-1} such that the high-intensity peak at 1320 cm^{-1} does not obscure examination.

Figure R15: a-b. Ratio of Raman mode intensities I_{293}/I_{441} (a) and I_{611}/I_{660} (b) for FL, CR ACR regions with (I) and without light (nl) as a function of bias. Error bars estimated by propagating uncertainty of peak intensity from experimental signal-to-noise-ratio.

Figure R16: a-c. Raw Raman spectra from T3 (a), T2 (b) and T1 (c) regions described in the main text, under bias dark/light (sub-550 nm wavelength of illumination over an area greater than $500 \mu\text{m}^2$). For the CR region the spectra have been averaged over 4 pixels to improve the signal-to-noise ratio and in the ACR region over 9 pixels. The spectra are truncated at 800 cm^{-1} such that the high-intensity peak at 1320 cm^{-1} does not obscure examination.

Figure R17: a-b. Ratio of Raman mode intensities I_{293}/I_{411} (a) and I_{611}/I_{660} (b) for T3, T2 and T1 regions with (I) and without light (nl) as a function of bias. Error bars estimated by propagating uncertainty of peak intensity from experimental signal-to-noise-ratio.

The following changes have in addition been made to the main text:

‘Repeating the above measurements under continuous illumination (400-550 nm; electrode pre-light soaked for 50 mins) shows limited difference in the response except in the FL regions (Supplementary Note 8). Intriguingly, a clear increase in the intensity ratio between E_g modes at 293 cm^{-1} and 411 cm^{-1} , linked to the degree of disorder and oxygen vacancies in $\alpha\text{-Fe}_2\text{O}_3$, is observed here^{66,67}. This tentatively suggests that light can accentuate structural (disorder) and vacancy rearrangement in ‘homogeneous’ regions of the sample under bias (see Supplementary Note 8 for further discussion).’

(2) In Figure 4a, the differential spectra demonstrated the order of oxygen vacancy concentration: $\text{ACR} > \text{FL} \approx \text{CR}$. However, the differential Raman spectra of CR and ACR in Figure 4c show a similarly negative trend, in contrast to that of FL. The different OER kinetics need to be considered and discussed.

We thank the reviewer for raising this interesting point, *i.e.*, why regions with seemingly different oxygen vacancy concentrations can show similar bias dependent Raman behaviour. Whilst our goal, has been to find correlations between the catalytic activity and local structure, getting a consensus between all techniques and the different information they access is challenging. Indeed, it is important to acknowledge this is a difficulty that has been found in purely ensemble spectroscopic studies of haematite where, for example, correlations have been observed between oxygen vacancies measured by EDX and only certain very specific Raman features [ACS Appl. Mater. Interfaces 2022, 14, 6615–6624, Chem. Sci., 2020, 11, 1085]. Hence, as the reviewer highlights, care is required in drawing conclusions.

The intensity and width of the E_g symmetry Raman modes of haematite have been shown to be related to OV. However, as we highlight in our text these modes also report on the Fe-O bond stretch and O-O bond stretches and more generally static and dynamic disorder, that may or may not be related to OVs [Nanoscale Res. Lett. 1, (2020), Chem. Mater. 21, 3763 (2009), ACS

Appl. Mater. Interfaces 2, 2804–2812 (2010), *J. Raman Spectrosc.* 23, 873–878 (1997)]. This contrasts with our differential reflectivity measurements which have been shown to be robustly linked exclusively to the OV concentration [*Proc. Natl. Ac. Sci.* 2012 109 (39) 15640-15645]. Hence, we use Raman to characterise more general disorder (which OVs fall under) and impurity phases such as FeOOH and reserve our reflectivity measurements for quantifying OV population specifically. Indeed, even though we can use Raman mode ratios to somewhat normalise for effects of sample thickness or laser penetration, in such inhomogeneous samples Raman is only semi-quantitative measure *i.e.*, changes in Raman intensity/ratios reflect changes in the local structure (including OVs) but do not necessarily say if there is more or less OVs simply from their magnitude [*ACS Appl. Mater. Interfaces* 2022, 14, 6615–6624, *Chem. Sci.*, 2020, 11, 1085]. It is for this reason we are cautious in our previous answer to the question of the reviewer when for example comparing the I_{293}/I_{441} Raman mode ratio to [OV] because this has only been shown semi-quantitatively and could be influenced by other things, compared to the reflectivity.

Considering this it is potentially not unexpected that the bias dependent behaviour of the Raman and reflectivity are not similar. The latter is dominated by only the properties of OVs but the former, as we state, is more closely linked to overall disorder and structural changes occurring during the OER mechanism. This might include changes in the length and electron density in Fe–O and O–O bonds, as the bias is raised and the formation of Fe(IV)=O [*Nat. Chem.* 12, 82–89 (2020) and *Nat. Chem.* 8, 778–783 (2016)]. These may not involve the OV population significantly resulting in distinct behaviour. Further theoretical and experimental work *e.g.*, measurements on epitaxial samples is required to understand the full effects of the bias dependent Raman behaviour. This is beyond the scope and message of this paper, where the goal has been to demonstrate that there are local structural changes during the photoelectrocatalytic OER and that these can vary drastically with micron scale morphology. **Nonetheless, to make clearer how we interpret the bias dependent Raman imaging, it's limitations and generally the differences between the data accessed by Raman and reflectivity measurements we have reworded our text as follows:**

‘The oxygen vacancy properties have been linked to the width and intensity of E_g symmetry peaks^{64,65} (specifically the intensity ratio between 293 cm^{-1} and 411 cm^{-1} modes^{66,67}, although there remains some debate as to which Raman features correlate most robustly with OVs/disorder in haematite).

...

We note that the Raman spectra do not report only on OVs, potentially explaining why the difference in behaviour between FL and CR/ACR is not observed in the differential reflectivity.’

Secondly, we agree that our discussion of OER kinetics is mostly based on previous literature understanding at the ensemble level on larger surfaces (areas of at least $1\times 1\text{ cm}^2$). For example, to interpret our local kinetics we lean on ref 37 of the original submission [*J. Phys. Chem. Lett.* 2020, 11, 7285–7290], where it was concluded that while oxygen vacancies have an important role in recombination losses, and the OER kinetics depend on the population of surface holes (as we discussed in Comment #2), the OER *mechanism* does not change, regardless the amount of oxygen vacancies or their filling state. We acknowledge that further (experimental and theoretical) studies are required to fully confirm the extension of such conclusions at the mechanistic level to the microscopic differences we are observing. Indeed, we believe that investigating the OER mechanism itself, *e.g.*, by trying to measure nanoscale rate laws, would

be a next step in more generally understanding the role of inhomogeneities. Thus far most kinetic analyses of haematite (and other thin-film photocatalytic oxides) in the literature focus on films with a preferential growth of the (110) crystal facet. In the films we study here many other facets can also be present. Hence, in the future by studying photocatalytic OER mechanisms independently on a range of facets we might be able to piece together an understanding of the mechanism in our films that contain many such facets.

However, we believe this step (which would first require the non-trivial preparation of films with only one type of facet) is well beyond the scope of this work. Indeed, something that complicates mechanistic understanding here, is our observation that the crystal structure may undergo some reorganisation during light driven OER (as reported by the community studying purely electrocatalytic materials, see *Nanoscale*, 2022,14, 15596-15606, *Angew. Chem.*, 2021, 60, 2561-2568 and *Nat. Mater.*, 2016, 15, 121-126 as examples).

In summary, linking the OER kinetics precisely to the different mechanisms at play is beyond the scope of our study. We have adjusted our text to note this and acknowledge that different exposed surfaces, displaying different surface energies, might influence the OER mechanism/reaction kinetics and further measurements are required to conclude on this.

We have modified the main text as follows:

*‘While the trends in FF and IPCE underscore the link between the local I-V curves, water oxidation efficiency and morphology we emphasise that our experiments do not allow us to comment on the OER mechanism (see **Supplementary Note 3** for further discussion).*

...

However, further developments will be needed especially to track defect dynamics over extended time periods and link thin-film morphologies to the OER mechanism itself. This could include the integration of microscopic transient photocurrent mapping (and an appropriate analysis framework⁶²), methods to quantify [OV] at sub-micron lengthscales, and using an LED for microscopic transient reflection measurements to better mimic solar irradiance⁷⁹. The light-driven migration/segregation of disorder (potentially OVs) we observe is intriguing but characterising the (local) oxide vacancy diffusion coefficient and activation barrier will be instrumental in learning how to exploit this phenomenon.’

The above discussion has also been included in the supporting information (Supplementary Note 3) on page 8.

Reviewer #3 (Remarks to the Author):

In this manuscript, Pandya et al. utilized a series of integrated microscopic mapping technologies to in situ study the activity–structure relationship in a prototypical hematite (α -Fe₂O₃) photoanode materials. Using a combined characterization of photoelectrochemical imaging, transient reflection microscopy and in situ Raman measurement alongside an X-ray absorbance measurement, the authors unveiled a significant spatial heterogeneity in activity and structure of α -Fe₂O₃, and found a positive correlation between photoelectrochemical activity and oxygen vacancy concentration. Considering that bulk-level studies are usually carried out in this field to reveal a link, the method proposed herein holds promises in promoting the development of highly efficient photoanode materials. On the other hand, this manuscript is not well written regarding the logic structure and languages. Therefore, I recommend this work to be published after major revisions listed below.

We thank the reviewer for highlighting our work as being novel and agreeing that the method we propose has significant scope and potential impact. We agree that the language and presentation of our manuscript needed work and we have extensively re-written the text/re-made figures to improve the work (alongside a host of new experiments to obtain further insights). We hope the reviewer will provide our revision suitably addresses their comments and is now suitable for publication in Nature Communications.

1. The title of this manuscript “correlating activity and defects in (photo)electrocatalysts using in-situ transient optical microscopy” can’t completely describe the study carried out in this work. Maybe a title of “correlating the activities and defect concentration in (photo)electrocatalysts using integrated in-situ microscopic imaging technologies” will be better.

We thank the reviewer for helping us better describe our work and agree the original title was only focussing on the transient microscopy we have retitled it to:

‘Correlating activities and defects in (photo)electrocatalysts using in-situ multi-modal microscopic imaging’

2. As for the steady-state reflectivity measurements of α -Fe₂O₃ shown in Figure 4a and 6b, the authors found that the most active region (ACR region or thinner area of T2/T3) possessed an interesting doublet signal, quite distinct from that of the lowly-active area. Have this phenomenon been reported in the case of α -Fe₂O₃ or other photoanode materials before? If using a transient reflection characterization, whether these double peaks will have same kinetics? How to completely understand the relationship between the electrochemical activity of α -Fe₂O₃ and such unique double peak signals? Maybe this is the most important findings obtained by integrated in-situ microscopic imaging technologies in this work. It will be better if the authors can strengthen this point.

We thank the reviewer for highlighting this point. We have provided an extensive answer to this question in question 3 of reviewer 2. For brevity (and to make this response more readable) we do not reproduce the answer here and direct the reviewer to the comments of reviewer 2. In summary we have performed new spectrally resolved transient absorption measurements, transient reflection microscopy at various probe wavelengths, polarisation dependent reflectivity measurements and electroabsorption to better characterise this feature. Whilst unequivocally assigning these peaks is exceptionally challenging, requiring further consensus

at the community level, both experimentally and theoretically, we believe the two states probed here are related to band-edge Fe *d-d* transitions. The high energy transition is potentially between a 6A_1 and 4A_1 state and the lower energy transition a 6A_1 and 4E state [*J. Phys. Chem. Sol.* 41, 9, 1980, 981-984]. The two peaks appear to have the same kinetics/electronics as well as chemical properties. Regions showing this double-peaked feature tend to show a higher photoelectrochemical activity but ascribing why exactly this is the case remains difficult, *i.e.*, we can only speculate how the additional transitions reported on by the double peak feature reduce back electron recombination/promote water oxidation/surface kinetics. Further work is needed to assign where exactly the double-peaks arises from and to confirm their impact in the OER mechanism and kinetics.

3. It seems that Figure 5c has a mistake. The presented Figure 5c is a photocurrent mapping rather than an EDX mapping result of T1, T2 and T3. Moreover, there seems a contradiction, maybe. The authors claimed that T1, T2 and T3 area had same Fe:O ration from EDX mapping (seemingly no differences in oxygen vacancies in these three areas). But on the other hand, the authors proposed that T1 area has more oxygen vacancies according to the results that T1 had bigger reflectivity change (Figure 6b). This suggested that the reflectivity change isn't only determined the concentration of oxygen vacancies, and also affected by the thinness of α -Fe₂O₃. In this sense, it is not solid to predict the oxygen vacancy concentration using steady-state reflectivity measurements (for example, Figure 4a).

We apologise for this error and causing confusion. **Figure 5c** was meant to be a photocurrent map as opposed to an EDX map. Furthermore, there was also an error in the analysis/labelling of our EDX spectra. What we in-fact find is that the oxygen content slightly drops on-going from T3 to T1 (as shown in **Figure R18**). This is consistent with the reflectivity measurements that suggest T1 also has a larger number of oxygen vacancies. This result is also somewhat in agreement with our assertion that T3 is similar to FL like regions. We consequently believe the correlation between reflectivity and vacancies remains robust. However, as we discuss above, in comment 6 of reviewer 1, some care is needed in quantifying the percentage of a given element from EDX, and the 'amount of oxygen' given by this number does not reflect purely oxygen vacancies. Whilst the differential reflectivity *vs* voltage spectra report on any kind of electronic transition promoted by UV-Vis photons that is sensitive to an applied potential, their potentiostatic evolution has previously been robustly assigned to oxygen vacancies, see *Chem. Sci.*, 2013,4, 2724-2734. We hence believe our analysis remains correct.

Figure R18: Oxygen content for T1, T2 and T3 regions estimated from EDX. Error on values is just below 0.05.

The reviewer makes an important point with regards to the thickness. Firstly, at the wavelength we measure the photocurrent locally, α -Fe₂O₃ has an absorption coefficient of $4 \times 10^4 \text{ cm}^{-1}$ (we note that this absorption coefficient is generally wavelength independent and is on the same order of magnitude for films synthesised using the routes in our work) [*Adv. Perf. Mat.* 33, 2018, 12]. Based on the Beer-Lambert law the light intensity will decay as shown in **Figure R19** inside the sample. The axial extent of our objective is 500 nm which means we must consider this decay in light intensity when calculating any quantities based on light absorption (namely the photocurrent and FF/IPCE). The 500 nm axial resolution of our objective prevents us from measuring the curve in **Figure R19** experimentally for our samples, but it has been reported also by other groups [*Nat. Commun.* 10, 4832 (2019)]. To normalise for this thickness variation between the T1, T2 and T3 regions considered in **Figure 5**, we can multiply the photocurrent measured at each voltage and spatial position by a pre-factor $\exp(\Delta z \times \alpha)$ where Δz is the difference in film thickness compared to the nominal film thickness of 300 nm and α the attenuation coefficient. In this way any trends in photocurrent we compare are free from thickness/light absorption effects. Importantly, following this correction our trends in the photocurrent and onset potential remain. We hence believe our observations remain robust.

Figure R19: Exponential decay of α -Fe₂O₃ absorption based on Beer-Lambert law.

Figure R20: Differential spectral reflectivity (with respect to reflection spectrum at $0.5 V_{RHE}$) as a function of bias from T1, T2 and T3, after correcting for all regions having a nominal thickness of 300 nm. The trends in the magnitude and shape of the reflectivity spectra remain the same after scaling.

With regards to the reflectivity measurements deriving any correction based on thickness is more challenging. The predominant reflection will be from the FTO/haematite interface based on the refractive index differences in our system and our objective focus position. While there will be some evanescent optical field into the material, the surface reflectivity we are measuring should in our view be independent of thickness. Indeed, we note that the magnitude of the $\Delta R/R$ signal follows a trend of $T3 > T2 > T1$ despite T3 being the thinnest region. Based on the above rationale we do not apply the above correction to the reflectivity spectra (or $\Delta R/R$ signals) but note that our trends would only be further accentuated by doing so (see **Figure R20**). As we do not draw a quantitative value on the oxygen vacancy concentration, as a result of the limitations detailed throughout this review, we do not believe our exact value is of importance. **Nonetheless, we have adjusted the main text of our manuscript as follows to account for the fact that our analysis and now explicitly consider the effect of different locations having different thicknesses:**

*‘Light absorption decays exponentially through the depth of the hematite film²⁸ following the Beer-Lambert law, becoming negligible below ~ 300 nm. Given this is the nominal thickness of our film (and the axial extent of the point spread function in our microscopy system is > 500 nm), to appropriately compare photocurrent densities and absorption between regions of different thickness, the signals must be scaled by the light-attenuation inside the material (see **Supplementary Note 11** for further details). Correcting for thickness variations, **Figure 5d** shows a map of the photocurrent density across T1, T2 and T3. The photocurrent density in T3 at $\sim 1.6 V_{RHE}$ is slightly larger ($\sim 1.27 \text{ mA cm}^{-2}$) than T1/T2 ($\sim 1.20 \text{ mA cm}^{-2}$); 15% uncertainty on values. This trend is also observed in the local fill-factors and IPCE values (FF: 0.33 (T3) and 0.19 (T1); IPCE: 18% (T3) vs 13% (T1)) which demonstrate the performance of T3 regions is on-par with high quality films⁷³. Finally, the photocurrent onset follows the order $T3 > T2 > T1$, i.e., potential for photocurrent generation and back electron transfer is lowest in T3.*

To assess the defect properties, specifically in terms of oxygen vacancies, we again perform biased spectral reflectivity measurements. As the reflection contrast predominantly arises from

the FTO/hematite interface, signals are not scaled by the sample thickness (but we note the same trends persist with/without scaling; see Supplementary Note 11)'

The above discussion has been added to pages 27-28 of the supporting information (Supplementary Note 11).

4. Though integrated in situ microscopic mapping, the author indicated a positive correlation between photoelectrochemical activity and oxygen vacancy concentration in α -Fe₂O₃ and provided the possibilities to tune the oxygen vacancy concentration via generating microscale cracks or reducing the thickness of α -Fe₂O₃. Can the author further give a quantitative description between photoelectrochemical activities of α -Fe₂O₃ and oxygen vacancy concentration? This is hard to be accessed in general bulk-level study but can really show the power and promise of the integrated in situ microscopic mapping in this field.

We agree with the Reviewer that it would be helpful to be as quantitative as possible. A similar question was posed in question 2 of reviewer 1 and for brevity (and legibility) the full answer provided there is not repeated. In summary using new local absorption measurements we have been able to characterise the local fill-factor at each microscopic position in the film, alongside the local I-V curves. These maps demonstrate further that the trends we observe do reflect genuine differences in activity across the samples. Quantifying the oxygen vacancy concentration is challenging. Whilst EDX could be an indicative measurement, as we detail in answer to question 6 of reviewer 1, it does not have the sensitivity or reliable quantitative nature to do so (see full answer there). Furthermore, the oxygen concentration that we could obtain from EDX does not exclusively relate to the vacancy concentration. This is because EDX is a compositional analysis technique with light elements being particularly difficult to quantify. Indeed, XPS is more commonly used to detect or quantify oxygen vacancy concentrations but this is challenging to achieve with the microscale resolution.

Whilst transient photocurrent measurements coupled with some impedance spectroscopy could be another technique used to perhaps quantify oxygen vacancy concentration, see *J. Am. Chem. Soc.* 2019, 141, 47, 18791–18798 or *Sol. RRL*, 2022, 6, 2200132, how to do this locally is unclear. (See answer to question 2 of Reviewer 2 why microscopic transient photocurrent measurements would struggle to provide meaningful information).

We direct the reviewer to the answers to questions 2 and 6 of reviewer 1 for the changes we have made to account for the above discussion.

5. Several other important literatures related to this manuscript should cited to help readers to better understand the background of this research (Fang et al., *Sci. Adv.* 2021, 7, eabj4452; Sadtler et. al., *J. Am. Chem. Soc.* 2021, 143, 11393–11403). And the most relevant work (i.e., Ref. 73, *Nature*, 2016, 530, 77–80) should be cited in the third paragraph of the “Main” part of this manuscript.

We thank the reviewer for highlighting these excellent works which use fluorescence microscopy to infer relationships between structure (defects, nanoparticle facets) and photocatalytic activity. **They have all been discussed/cited in the main and the text modified appropriately, as follows:**

‘Optical (super-resolution) fluorescence microscopies^{34–38} can somewhat alleviate the problem of in-situ probing and have unveiled the influence of inter-facet junctions^{39,40}, defect sites and

nanoparticle structure on photocatalytic activity in a host of systems (2D materials³⁵, oxides and zeolites¹⁷). But the reliance on indirect exogenous fluorescent probes makes this method difficult to apply to a wide range of photocatalytic systems.'

6. The readability and the language of this manuscript should be further improved, especially the part of in situ Raman and X-ray absorbance measurements. Maybe the authors should provide some schematic illustrations to correlate the Raman/X-ray characterization results with the photoelectrochemical activity of α -Fe₂O₃ to help readers to better understand these obtained structural information. Moreover, the manuscript should be checked more carefully. For instance, several obvious mistakes appear in the first paragraph of "(Photo)electrocatalysis at microstructural breaks in the film" section, such as "marked with red/green asterisk", "peaks at 0.71 eV/6.4 eV and 0.51 eV", "the Fe:O ratio in ACR regions".

We thank the reviewer for highlighting this point to us and helping us to improve the readability/presentation of our work. We fully agree with the reviewer and have extensively been through the manuscript rewording large parts to improve readability and language, as well as checking carefully for typos (including those pointed out by the reviewer).

We have further reworked Figure 1 (and all captions in general) to provide more information about the exact information obtained from each technique and how these pieces of information correlate. Whilst we did try to merge Figure 1 (which provides a schematic of the techniques we use) with Figure 2 (the information obtained/correlations) we found this led to a more confusing figure overall. Hence, we have kept these figures separate and instead improved our descriptions and language to be clear on how to correlate the Raman/X-ray/reflectivity characterisation results with the photoelectrochemical activity of α -Fe₂O₃, such that readers can better understand the obtained information. We believe the two schematics in Figure 1 and 2 together do provide a good illustration of the general method we have developed and the information that can be obtained. Given the vast and detailed amount of imaging data we have obtained, we have reserved the remainder of the Figures for experimental data as opposed to providing illustrations, relying on the reworked text for clearer explanation. While further sub-headings might potentially help, we understand these to be against the journal formatting policy. In our view the more extensive supporting information and methods section we provide, alongside a heavily reworked text (with new experiments) greatly improve the readability and understanding of our work to a broad audience of readers.

REVIEWERS' COMMENTS

Reviewer #2 (Remarks to the Author):

All my concerns have been well addressed. The manuscript can be accepted in its present form.

Reviewer #3 (Remarks to the Author):

The authors have devoted considerable efforts to revise their manuscript according to my suggestions and those of other reviewers. After carefully reviewing this work and their responses again, I think this work can now potentially be published in Nature Communications after a further citation of two important papers on pump-probe characterization of semiconductor carriers (Wang et al., Adv. Funct. Mater. 2022, 32, 2107551; Xu et al., J. Am. Chem. Soc. 2022, 144, 13928–13937).

Reviewer #5 (Remarks to the Author):

The main result of the paper is the development of a multimodal correlated optical microscope (for in-situ measurement of photocurrent, reflectivity, ns to ms transient reflection microscopy, and Raman). The advances and limitations of this approach are exemplified by the study a photoelectrode model system (hematite α -Fe₂O₃ photoanodes). Although the confirmation of microstructure-performance correlations for this particular system are not particularly surprising, the technique has potential to contribute to in-situ microstructural characterisation of active photoelectrodes in general.

The authors have done a major revision of their work considering the feedback of the referees. They have included an extensive literature discussion and provided the most plausible data interpretation, while recognising the limitations of both the main approach, and the supporting techniques. I consider that they have properly addressed the applications and limitations of their approach, and added the relevant information to analyse the data.

From my perspective the work is sound and ready to be published either in Nature Communications, or in a more specialised spectroscopy journal. Publishing it here will give the technique visibility to other communities, which might allow further refining of the technique and applications in a variety of systems.

We thank the reviewers for their time taken to review and improve our work. Below we have addressed the remaining comments (blue) with our answers (and any text modifications) in black and red, respectively.

Reviewer #2 (Remarks to the Author):

All my concerns have been well addressed. The manuscript can be accepted in its present form.

We thank the reviewer for their time taken in reviewing and improving our work. We are pleased that it is now suitable for publication.

Reviewer #3 (Remarks to the Author):

The authors have devoted considerable efforts to revise their manuscript according to my suggestions and those of other reviewers. After carefully reviewing this work and their responses again, I think this work can now potentially be published in Nature Communications after a further citation of two important papers on pump–probe characterization of semiconductor carriers (Wang et al., Adv. Funct. Mater. 2022, 32, 2107551; Xu et al., J. Am. Chem. Soc. 2022, 144, 13928–13937).

We thank the reviewer for their time taken in reviewing and improving our work. We are pleased that it is now suitable for publication. The two important references the reviewer directs us to have now been cited at the following locations in the text (references marked in bold and underlined below):

‘Optical (super-resolution) fluorescence microscopies^{34–38} can somewhat alleviate the problem of in-situ probing and have unveiled the influence of inter-facet junctions^{39,40}, defect sites and nanoparticle structure on photocatalytic activity in a host of systems (2D materials^{35,41}, oxides and zeolites¹⁷).

...

In this work, we develop a multi-modal microscopy setup encompassing ns to ms transient reflection imaging^{42,43}, Raman imaging, and local (spectro)electrochemistry, to map sub-micron electrochemical dynamics in photoelectrocatalysts.’

Reviewer #5 (Remarks to the Author):

The main result of the paper is the development of a multimodal correlated optical microscope (for in-situ measurement of photocurrent, reflectivity, ns to ms transient reflection microscopy, and Raman). The advances and limitations of this approach are exemplified by the study a photoelectrode model system (hematite α -Fe₂O₃ photoanodes). Although the confirmation of microstructure-performance correlations for this particular system are not particularly surprising, the technique has potential to contribute to in-situ microstructural characterisation of active photoelectrodes in general.

The authors have done a major revision of their work considering the feedback of the referees. They have included an extensive literature discussion and provided the most plausible data interpretation, while recognising the limitations of both the main approach, and the supporting techniques. I consider that they have properly addressed the applications and limitations of their approach, and added the relevant information to analyse the data.

From my perspective the work is sound and ready to be published either in Nature Communications, or in a more specialised spectroscopy journal. Publishing it here will give the technique visibility to other communities, which might allow further refining of the technique and applications in a variety of systems.

We thank the reviewer for their time taken in reviewing our work. We also hope that the methods will be taken forward in the community to develop further insights into the role of microstructure in photoelectrocatalyst performance.